# Trust Your Robots! Predictive Uncertainty Estimation of Neural Networks with Sparse Gaussian Processes

**Jongseok Lee**[1,2] **Jianxiang Feng**[1,3] **Matthias Humt**[1,3] **Marcus G. Müller**[1,4] **Rudolph Triebel**[1,3]
[1]Institute of Robotics and Mechatronics, German Aerospace Center (DLR)
[2]High Performance Humanoid Technologies, Karlsruhe Institute of Technology (KIT)
[3]Chair of Computer Vision and Artificial Intelligence, Technical University of Munich (TUM)
[4]Autonomous Systems Laboratory, ETH Zürich (ETHZ)

**Abstract:** This paper presents a probabilistic framework to obtain both reliable and fast uncertainty estimates for predictions with Deep Neural Networks (DNNs). Our main contribution is a practical and principled combination of DNNs with sparse Gaussian Processes (GPs). We prove theoretically that DNNs can be seen as a special case of sparse GPs, namely mixtures of GP experts (MoE-GP), and we devise a learning algorithm that brings the derived theory into practice. In experiments from two different robotic tasks – inverse dynamics of a manipulator and object detection on a micro-aerial vehicle (MAV) – we show the effectiveness of our approach in terms of predictive uncertainty, improved scalability, and run-time efficiency on a Jetson TX2. We thus argue that our approach can pave the way towards reliable and fast robot learning systems with uncertainty awareness.

**Keywords:** Robotic Introspection, Bayesian Deep Learning, Gaussian Processes

## 1 Introduction

This work aims to provide an algorithm that can estimate uncertainty of DNN predictions reliably and fast, and at the same time, is suited for integration into a large range of robotic systems. Generic solutions to this problem are crucial for safe robot learning and introspection [1, 2], giving the robots with an ability to assess their own failure probabilities and to alter their behaviors towards safety. While the state of the art is advancing [3, 4, 5], we still face practical difficulties in two main domains. First, the current methods are often not efficient at test-time, e.g. for a single input, the methods require multiple predictions from several copies of a model [4] or samples from the model's distribution [3]. And second, in general these methods do not provide reliable uncertainty estimates when compared to GPs - often known as the golden standard of probabilistic machine learning [6, 7].

Therefore, we propose to estimate the *predictive uncertainty* of DNNs using MoE-GPs [8, 9] - a sparse variant of GPs that divides the input space into smaller local regions using a *gating function*, where individual GPs called *experts* learn and make predictions (see figure 1). To do so, we provide both theoretic foundations and a practical learning algorithm. First, we formally derive a connection between DNNs and MoE-GPs. As a result, we reveal how MoE-GPs with a DNN-based kernel [10] can provably approximate uncertainty in DNNs. Moreover, we devise a learning algorithm that brings the derived theory into practice. Our solution involves a gating function that strictly divides the input data into smaller subsets by performing clustering in kernel space, and we propose the concept of a patchwork prior, mitigating the problem of discontinuity between local GP experts at their boundaries. For efficiency, we further propose to exploit active learning and model compression techniques.

Our approach has several favorable features for many classification and regressions tasks in robotics. At run-time, it maintains the predictive power of a DNN, and its uncertainty estimates do not require combining multiple predictions of DNNs. Moreover, we inherit the benefits of MoE-GPs for uncertainty estimates. These include an improved scalability when compared to a GP with the neural tangent kernel [10] (NTK), as well as natural support for distributed training. We contend that such probabilistic methods must run efficiently on a robot and as applicable as the sparse GPs. Therefore, we not only provide ablation studies and evaluations against the state-of-the-art (SOTA), but we also show that our method (i) scales to approximately 2 million data points for the task of learning inverse

5th Conference on Robot Learning (CoRL 2021), London, UK.

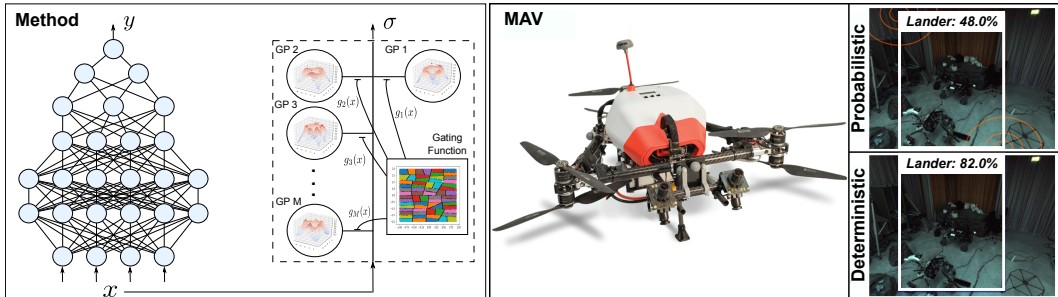

Figure 1: Our method (left) computes uncertainty $\sigma$ of Neural Network predictions **y** using MoE-GPs with the Neural Tangent Kernels (NTK). With this, a MAV [11] (right) performs probabilistic object detection at an interactive frame rate, reducing overconfidence of an object detector. Only a feedforward network is shown but our method also applies to convolution and recurrent DNNs.

dynamics, and (ii) on a Jetson TX2 of a MAV, runs more than 12 times faster than the widely used MC-dropout [3], while performing object detection within a scenario for planetary exploration.

**Contributions** In summary, our main contribution is a novel method for estimating predictive uncertainty of DNNs with sparse GPs (section 3.1), backed up by (i) a theoretical connection between DNNs and MoE-GPs (section 3.2), (ii) a learning algorithm for its applicability in practice (section 3.3), and (iii) an exhaustive empirical evaluation that shows the benefits of our approach (section 4).

## 2 Related Work

Many researchers explored the idea of robotic introspection [2], i.e. robots that reason about *"when they don't know"* in order to avoid catastrophic failures. So far, many proposed methods used the robots' past experiences of failures [12, 13, 14], or detect the inconsistencies within the predictors [15, 16]. These works provide strong evidence that introspection improves the reliability of the robots. Other methods are based on uncertainty estimation [1], where learning algorithms such as DNNs use probability theory to express their own reliability [17, 18, 19]. Commonly, these works use a technique called MC-dropout [3]. However, the intense computational burden in estimating uncertainty via sampling or ensembles [20] limits their applicability in real-time systems [11, 21, 22].

Several techniques without sampling or ensembles have been presented for run-time efficiency. These methods often use either Dirichlet distributions or functional Bayesian Neural Networks [23, 24]. Unfortunately, the applicability of Dirichlet distributions is limited to classification, while functional Bayesian Neural Networks face hurdles in scaling to large datasets without approximations [24]. We recommend the recent works of He et al. [25] and Adlam et al. [26], who use NTK-based GPs for uncertainty estimation of DNNs and provide useful insights. However, we focus on a technique that can be readily deployed on a real-time system such as a MAV. Hence, we take a different path to ensembles of infinite width DNNs [25, 26]. We find uncertainty propagation techniques [27, 28, 29] to be practical as these methods are training-free, i.e. assume existing and already well-trained DNNs.

In addition, our theoretic contribution (section 3.2) extends on how DNNs are related to GPs (pioneered by Neal [30] and others [31, 10, 32]), where we derive a relationship between DNNs and MoE-GPs as opposed to a single GP. Theoretic insights that connect DNNs and sparse GPs may pave the way towards their application, as GPs alone do not scale to big data required by DNNs. Lastly, we extend previous works on MoE-GPs (section 3.3). MoE-GPs are efficient but suffer from non-smooth uncertainty estimates [33]. Park and Apley [34] show an elegant solution called patchwork kriging. By augmenting data at the boundaries between GP experts, the experts are forced to produce similar predictions at their boundaries. Yet, as patchwork kriging is limited to low-dimensional inputs [34], we propose several techniques that extend it to higher dimensions, like images (assuming the NTK).

## 3 Materials and Methods

We describe a method that addresses the problem of quantifying uncertainty estimates in DNN predictions, without sampling or ensembles. First, we introduce the main concept of estimating predictive uncertainty with sparse GPs, followed by theoretic results and our learning algorithm.

## 3.1 Main Idea: Fast Uncertainty Estimation with Sparse Gaussian Processes

Figure 1 visualizes our approach. The proposed predictive model uses an existing, and already well-trained DNN for accurate predictions. At the same time, the method offers the possibility to obtain the predictors' reliable uncertainty in a closed form solution, using a DNN-based MoE-GPs. Intuitively, we can linearize and transform DNNs around a mode to obtain a linear model. As any linear models are GP regressors from a function space view, we can get a GP-based representation of DNNs with a DNN-based kernel, the NTK. We describe below how the obtained GPs can estimate the uncertainty of DNNs, but cannot replace the DNN predictions. Moreover, as full GPs do not scale to big data, we choose to use MoE-GPs, which divides the data into smaller subsets and form many smaller GPs per these subsets. This results in the proposed combination of DNNs and sparse GPs.

**Fundamentals** First, we introduce our notation. Considering a supervised learning task on input-output pairs $\mathcal{D} = \left\{ \mathcal{X}, \mathcal{Y} \right\} = \left\{ (\boldsymbol{x}_i, \boldsymbol{y}_i) \right\}_{i=1}^{N}$, where $\boldsymbol{x}_i \in \mathbb{R}^D$, $\boldsymbol{y}_i \in \mathbb{R}^K$, we describe a DNN as a parametrized function $f_{\boldsymbol{\theta}} : \mathbb{R}^D \to \mathbb{R}^K$, where $\boldsymbol{\theta} \in \mathbb{R}^P$. Here, learning typically seeks to obtain an empirical risk minimizer of the loss function, i.e. $\min_{\boldsymbol{\theta}} \quad \frac{1}{|\mathcal{D}|} \sum_{(\boldsymbol{x}, \boldsymbol{y}) \in \mathcal{D}} \mathcal{L}(f_{\boldsymbol{\theta}}(\boldsymbol{x}), \boldsymbol{y}) + \frac{\delta}{2} \boldsymbol{\theta}^T \boldsymbol{\theta}$ where $\delta$ is an $L_2$-regularizer, and mini-batches $\mathcal{B} \subset \mathcal{D}$ are used to find a local maximum-a-posteriori (MAP) solution $\hat{\boldsymbol{\theta}}$. We assume a twice differentiable and strictly convex loss function $\mathcal{L}$, e.g. mean squared error (MSE), and piece-wise linear activations in $f_{\boldsymbol{\theta}}$ (e.g. RELU). For a clear exposition, we drop the indices $i$.

To set the scene for the paper, the Neural Linear Models (NLMs) [27, 35, 36, 32] are defined, which can estimate a DNNs' predictive uncertainty. To do so, consider a transformed dataset $\widetilde{\mathcal{D}} = \left\{ \mathcal{X}, \widetilde{\mathcal{Y}} \right\}$ with the pseudo-output $\widetilde{\boldsymbol{y}} := \boldsymbol{J}_f(\boldsymbol{x})\hat{\boldsymbol{\theta}} - \boldsymbol{H}_{\mathcal{L}}(\boldsymbol{x}, \boldsymbol{y})^{-1} \boldsymbol{R}_{\mathcal{L}}(\boldsymbol{x}, \boldsymbol{y})$. This transformation uses the model derivatives of the underlying DNN around $\hat{\boldsymbol{\theta}}$, namely the Jacobian $\boldsymbol{J}_f(\boldsymbol{x}) := \partial f_{\boldsymbol{\theta}}(\boldsymbol{x})/\partial \boldsymbol{\theta}^T \in \mathbb{R}^{K \times P}$, the Hessian $\boldsymbol{H}_{\mathcal{L}}(\boldsymbol{x}, \boldsymbol{y}) := \partial^2 \mathcal{L}(f_{\boldsymbol{\theta}}(\boldsymbol{x}), \boldsymbol{y})/\partial f_{\boldsymbol{\theta}}(\boldsymbol{x})^T \partial f_{\boldsymbol{\theta}}(\boldsymbol{x}) \in \mathbb{R}^{K \times K}$ and the residuals $\boldsymbol{R}_{\mathcal{L}}(\boldsymbol{x}, \boldsymbol{y}) := \partial \mathcal{L}(f_{\boldsymbol{\theta}}(\boldsymbol{x}), \boldsymbol{y})/\partial f_{\boldsymbol{\theta}}(\boldsymbol{x}) \in \mathbb{R}^K$. Assuming a white noise and an uninformative prior, the NLMs are:

$$\widetilde{\boldsymbol{y}} = \boldsymbol{J}_f(\boldsymbol{x})\boldsymbol{\theta} + \epsilon \quad \text{with} \quad \epsilon \sim \mathcal{N}(\boldsymbol{0}, \boldsymbol{H}_{\mathcal{L}}(\boldsymbol{x}, \boldsymbol{y})^{-1}) \quad \text{and} \quad p(\boldsymbol{\theta}) \sim \mathcal{N}(\boldsymbol{0}, \delta^{-1}\boldsymbol{I}). \tag{1}$$

The NLMs can be thought as a Bayesian linear model with learned features from a DNN, which is obtained via a linearization around the DNNs' last layer [27]. Thus, the isotropic prior is governed by the parameter $\delta$ that corresponds to the $L_2$ regularizer [37] while the white noise is governed by the term $\boldsymbol{H}_{\mathcal{L}}^{-1} \boldsymbol{R}_{\mathcal{L}}$ in the described data transformation, given that the DNN fit the train data well, e.g. $\boldsymbol{R}_{\mathcal{L}} \approx 0$. The term $\boldsymbol{H}_{\mathcal{L}}$ is independent of the input if the 2nd derivative of a loss is a constant, e.g. MSE. Moreover, for commonly used loss functions such as cross entropy and MSE, the covariance $\boldsymbol{\Sigma}$ of DNN predictions $f_{\boldsymbol{\theta}}$ is equal to that of the NLMs' $\widetilde{\boldsymbol{y}}$ [27, 32]. This means that for the same input $\boldsymbol{x}$, the NLMs can estimate the predictive uncertainty of DNNs. Yet, $\widetilde{\boldsymbol{y}}$ cannot be used to replace $f_{\boldsymbol{\theta}}$, due to their differences in $f_{\boldsymbol{\theta}}$ and $\widetilde{\boldsymbol{y}}$. In the Appendix, we provide concrete examples and their derivations.

**Main idea** Let's now apply a kernel trick [38] to the NLMs, so that we can obtain an effective combination of DNNs with GPs. To explain, we can further map the NLM to a function space, as opposed to working in the weight space. Doing so, a well known function space formulation of linear models [38] turns the NLM into a GP with the kernel $\boldsymbol{K} = \frac{1}{\delta} \boldsymbol{J}_f(\boldsymbol{x})^T \boldsymbol{J}_f(\boldsymbol{x})$, which is also known as the *Network Tangent Kernel* (NTK). Thus, GPs with NTK can be used to estimate the predictive uncertainty of DNNs, as their equivalent NLMs can. Here, what motivates the formulation is the idea of deep kernel machines [39], i.e. we learn the kernel representations as oppose to hand designing the features and the kernels, and in our case, the kernel representations are the tangents of DNNs' Jacobians. Importantly, doing so, we get a training-free combination of DNNs with GPs, as we keep DNN predictions the same as the MAP, while estimating their uncertainty using GPs with NTK.

Unfortunately, such combinations of DNNs with GPs are restricted by a prominent weakness of standard GPs [40]: the cubic time complexity $O(N^3)$ that grows with the dataset size $N$. So, the computational costs are prohibitive for large scale problems that DNNs assume (typically referred to $N > 10000$ in [40, 38]). While we leave the theoretic motivation to section 3.2, we thus propose MoE-GPs with NTK, in order to advance the scalability of the proposed combination. A MoE-GP consists of $M$ experts of GPs or learners $\mathcal{F} = \left\{ \widetilde{f}_{\text{GP}_1}, \cdots, \widetilde{f}_{\text{GP}_M} \right\}$, and a gating function $g : \mathbb{R}^D \to \Delta^{M-1}$ that maps any input $\boldsymbol{x}$ to $g(\boldsymbol{x}) = [g_1(\boldsymbol{x}), \cdots, g_M(\boldsymbol{x})]$ [8, 9]. Each expert of the MoE learns and predicts within a subset of the input domain, and a gating function generates these subsets.

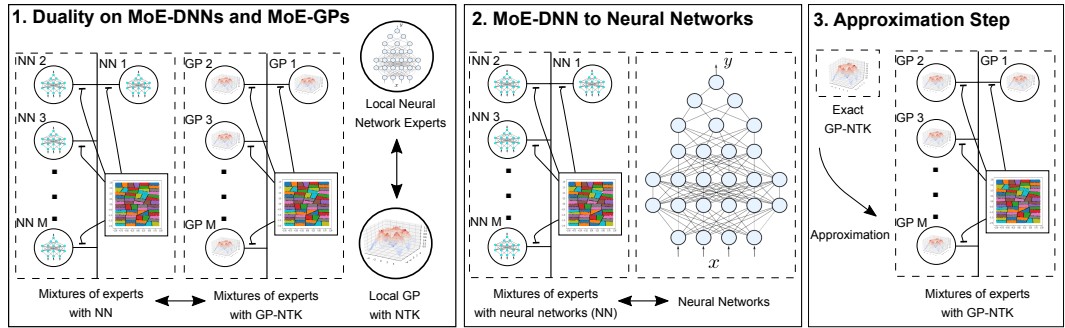

Figure 2: We visualize theoretic results as a roadmap of the proof path. Detailed theorems and their proofs are in the Appendix. (1) Assuming a strict division of data, a duality between mixtures of experts (MoE) with DNNs and GPs are established first. (2) To expand the applicability of the results beyond MoE-DNN, we show that the input-prediction relationships of a single DNN and a MoE-DNN are equivalent with an assumption on the identical local DNN experts. (3) This results in a provable approximation error between the proposed MoE-GPs and the true GPs with the NTK.

Specifically, we choose a gating function $g_m(\boldsymbol{x}) = 1$ in just one coordinate for each input [41]. Such strict partition enables the use of a local model for each input [34], avoiding the combinations of multiple predictions of a model for each input, e.g. model averaging as in Meier et al. [42]. The subscripts $m = 1, 2, ..., M$ denote the $m^{\text{th}}$ expert throughout the paper. Then, we write a MoE-GP as,

$$\widetilde{\boldsymbol{y}} = \sum_{m=1}^{M} g_m(\boldsymbol{x})\widetilde{f}_{\text{GP}_m}(\boldsymbol{x}) + \epsilon_m \quad \text{with} \quad \widetilde{f}_{\text{GP}_m}(\boldsymbol{x}) \sim \text{GP}(\boldsymbol{0}, \frac{1}{\delta_m}\boldsymbol{J}_{f_m}(\boldsymbol{x})^T\boldsymbol{J}_{f_m}(\boldsymbol{x})). \tag{2}$$

As depicted in figure 1, the gating function $g_m(\boldsymbol{x})$ assigns the input data $\boldsymbol{x}$ to the $m^{\text{th}}$ subset, and individual GPs $\widetilde{f}_{\text{GP}_m}(\boldsymbol{x})$ learn and make predictions for the assigned data within the $m^{\text{th}}$ subset. Consequently, the generative model and the predictions $\mathcal{N}(\widetilde{\boldsymbol{y}}_m^*, \boldsymbol{\Sigma}_m(\boldsymbol{x}^*))$ on the new test datum $\boldsymbol{x}^*$ are:

$$\begin{bmatrix} \widetilde{\boldsymbol{y}}_m \\ \widetilde{f}_{\text{GP}_m} \end{bmatrix} \sim \mathcal{N}\left(\boldsymbol{0}, \begin{bmatrix} \boldsymbol{K}_m + \sigma_{0,m}\boldsymbol{I} & \boldsymbol{k}_{m,*} \\ \boldsymbol{k}_{m,*}^T & \boldsymbol{k}_{m,**} \end{bmatrix}\right) \quad \text{and} \quad \begin{aligned} \widetilde{\boldsymbol{y}}_m^* &= \boldsymbol{k}_{m,*}^T(\boldsymbol{K}_m + \sigma_{0,m}\boldsymbol{I})^{-1}\widetilde{\boldsymbol{y}}_m, \\ \boldsymbol{\Sigma}_m &= \boldsymbol{k}_{m,**} - \boldsymbol{k}_{m,*}^T(\boldsymbol{K}_m + \sigma_{0,m}\boldsymbol{I})^{-1}\boldsymbol{k}_{m,*} + \sigma_{0,m}, \end{aligned}$$

where, $\boldsymbol{K}_m = \boldsymbol{K}_m(\mathcal{X}, \mathcal{X})$, $\boldsymbol{k}_{m,*} = \boldsymbol{k}_m(\mathcal{X}, \boldsymbol{x}^*)$ and $\boldsymbol{k}_{m,**} = \boldsymbol{k}_m(\boldsymbol{x}^*, \boldsymbol{x}^*)$. This posterior predictive distribution $\boldsymbol{\Sigma}_m$ is an indicator of total uncertainty, i.e. the kernel function captures the model uncertainty while the constant term $\sigma_{0,m}\boldsymbol{I}$ corresponds to the aleatoric uncertainty [43]. In GPs, as we do not have access to exact function values, the aleatoric uncertainty often relies on the assumption of additive i.i.d white noise, and in MoE-GPs, the terms $\sigma_{0,m}\boldsymbol{I}$ are inferred from the data per subset, leading to a non-stationary kernel. This is achieved by optimizing the marginal likelihood [38].

The benefits of the MoE-GP formulation are two-folds. First, as shown above, the generative model of a GP is defined per subset, and therefore, the GP experts are smaller than the one on the entire dataset. This improves the computational complexity of GPs [34]. Second, the covariance computations are in a closed form. Thus, we neither need sampling nor ensemble [3, 4]. Using the Lanczos approximation [44], which approximates the matrix inversion with the multiplications (suited for GPUs), we can compute the uncertainty estimates from GPs, with a constant time complexity [44].

For classification, we leverage the framework of Lu et al. [45], which can perform classification with the regression models via confidence calibration [5]. While we refer to Lu et al. [45] for the details, this steps enables us to obtain the classification uncertainty in closed form, i.e, for a class $c$:

$$p(c|\boldsymbol{z}_m) = \text{softmax}\left(\frac{\boldsymbol{z}_m}{T_m}\right) \text{ with } T_m = \sqrt{1 + \lambda_{m,0}\boldsymbol{\Sigma}_m(\boldsymbol{x}^*)}, \tag{3}$$

where $\boldsymbol{z}_m$ is the activation, $T_m$ is the temperature scaling factor, which is a function of a scaling constant $\lambda_{m,0}$, and GP regression uncertainty $\boldsymbol{\Sigma}_m(\boldsymbol{x}^*)$. Intuitively, high prediction uncertainty will reduce the corresponding $p(c|\boldsymbol{z}_m)$ by increasing the $T_m$, thereby calibrating the confidences [5, 46, 45].

### 3.2 On Theory: Neural Networks as Sparse Gaussian Processes

So far, we have outlined our main idea - using the function space view of NLMs and dividing the input space into smaller subsets, we can form smaller GPs per division. These GPs then estimate the

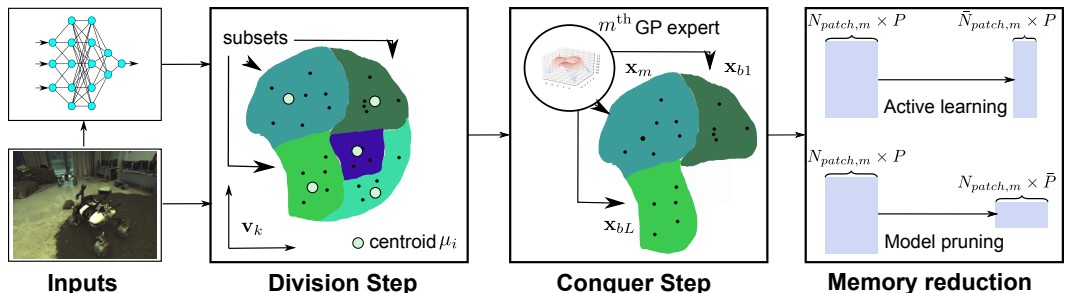

Figure 3: The proposed algorithm involves (i) the division of input data using Jacobians of already trained DNNs, (ii) training of GP experts within their partitions and neighboring regimes, and (iii) combining active learning and model pruning techniques to reduce space complexity.

predictive uncertainty of DNNs. Now, we discuss the theoretical foundations on how DNNs can be cast as MoE-GPs with NTK. Due to space constraints, we refer the reader to the Appendix for in depth treatment, where we provide various background materials, our theorems and their proofs. In section 3.3 we explain our learning algorithm while here, we briefly summarize the main results.

**Main result** Figure 2 summarizes our findings. Consider a mixtures of experts model, where the experts are DNNs and a gating function strictly divides the input space, i.e only one local DNN expert per division (MoE-DNN). Then, instead of the MAP parameter estimates $\hat{\boldsymbol{\theta}}$, we consider Bayesian DNNs with Gaussian distributions $p(\boldsymbol{\theta}|\mathcal{D})$, i.e representing DNNs as probability distributions over their parameters (obtained using [47, 48, 49, 50]). As a first step, as a specific instance of Khan et al. [32], we show that the DNN experts have mathematically equivalent Gaussian distributions with GPs using NTK. We further show that MoE-DNNs and MoE-GPs have equivalent Gaussian distributions, establishing the duality of the two models in a Bayesian sense. Main insight to the former is the probabilistic independence between the local experts due to a strict division of the input space.

Next, we establish a connection between a DNN, and a MoE-GP as shown in figure 2, in order to increase the applicability of our theoretic results. To do so, we point out that the input-prediction relationships of a single DNN and a MoE-DNN are equivalent, if all the local DNN experts are the same as the single DNN. Under the given conditions, we derive that a single DNN can be cast as a MoE-GP, with an assumption that the data is stationary. The resulting step yields the approximation error of $\|\boldsymbol{K}(\mathcal{X},\mathcal{X}) - \boldsymbol{K}_{true}(\mathcal{X},\mathcal{X})\|_F^2 = \sum_{ij} \boldsymbol{K}(\boldsymbol{x}_i, \boldsymbol{x}_j)^2 - \sum_{m=1}^M \sum_{ij \in \mathfrak{V}_m} \boldsymbol{K}(\boldsymbol{x}_i, \boldsymbol{x}_j)^2$ where $\boldsymbol{K}_1 = \boldsymbol{K}_2 = \cdots = \boldsymbol{K}_M$ for all $M$ GP experts (while it is still possible to keep different hyper-parameters $\delta_m$ and $\sigma_m$), and $\boldsymbol{K}_{true}$ is the kernel matrix of the true DNN-equivalent GP with NTK. This means that less approximation error occurs when less correlated data points are assumed to be independent by MoE-GP, while strongly correlated points by NTK are within the same GP experts.

### 3.3 From Theory to Practice: A Practical Learning Algorithm

Now, we attempt to bring our theoretical results into a practical tool for robotics. As shown in figure 3 our solution involves the division of data into several partitions. Then, in a conquer step, we train GP experts within the partitions. Moreover, we propose concepts to reduce the memory complexity.

**Division step** We design a gating function that divides the input data into smaller regimes s.t. highly correlated data points are grouped together whilst the points with weak correlations are separated apart. To do so, we perform NTK PCA, i.e. kernel PCA that uses NTK to reduce the dimensions of the data. Then, K-means is applied to form the clusters. Concretely, NTK PCA reduces the input data $\boldsymbol{x}_i$ for $i = 1, ..., N$ into the low dimensional principal components $\boldsymbol{v}_k = \sum_{i=1}^N \alpha_{ik} \boldsymbol{J}_f(\boldsymbol{x})^T \boldsymbol{J}_f(\boldsymbol{x}_i)$, where $\boldsymbol{\alpha}$ is the eigenvector of $\widetilde{\boldsymbol{K}} \boldsymbol{\alpha}_k = \lambda_k \boldsymbol{\alpha}_k$ with a centered kernel matrix $\widetilde{\boldsymbol{K}} = \boldsymbol{K} - 2\mathbf{1}_N \boldsymbol{K} + \mathbf{1}_N \boldsymbol{K} \mathbf{1}_N$ for the matrix of ones $\mathbf{1}_N \in \mathbb{R}^{N \times N}$. Then, the centroid $\mu_i$ and division labels $\boldsymbol{c}$ are computed, iterating:

$$\boldsymbol{c}_i = \arg\min \sum_i^M \sum_{\boldsymbol{v}} \|\boldsymbol{v}_i - \mu_i\|^2 \quad \text{and} \quad \mu_i = \frac{\sum_i^M \mathbf{1}\{\boldsymbol{c}_i = i\} \boldsymbol{v}_i}{\sum_i^M \mathbf{1}\{\boldsymbol{c}_i = i\}}. \tag{4}$$

The proposed technique gives a solution for Kernel K-means [51], which minimizes the derived error bound $\|\boldsymbol{K}(\mathcal{X},\mathcal{X}) - \boldsymbol{K}_{true}(\mathcal{X},\mathcal{X})\|_F^2$ (section 3.2) with a balancing normalization [52]. As kernel PCA

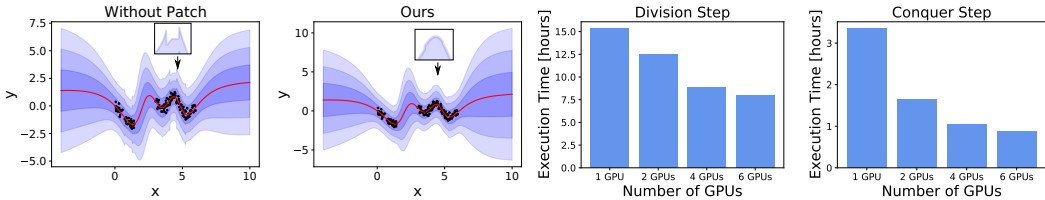

Figure 4: (Left) We visualize predictive uncertainty of DNNs with and without the proposed patch-work prior. The black dots are train data points, and the red line shows the predictions of a deterministic network. Blue shades show up to three standard deviations. Without the patch prior, sharp peaks are observed in uncertainty estimates. (Right) Training time for approximately 2 million data points is shown for a varying the number of GPUs. We learned 512 GP experts per joint torques.

does not scale to large datasets, we use 2-step kernel K-means [53], which uses randomly sampled and less data to compute the cluster centroids. We refer to Chitta et al. [53] for more details.

**Conquer step** Next, we form the individual GP experts per divided local regime. A naive strategy is to form a single GP per regime. Unfortunately, such a model suffers from discontinuity in predictions at the boundaries [33]. For example, input points at the cluster boundaries may yield different predictions from the surrounding GPs, causing sharp peaks in predictions. Therefore, we propose:

$$\boldsymbol{K}_{m,\text{patch}} = \left[\boldsymbol{J}_{f_m}(\boldsymbol{x}_m), \boldsymbol{J}_{f_mb1}(\boldsymbol{x}_{b1}), \cdots \boldsymbol{J}_{f_mbL}(\boldsymbol{x}_{bL})\right]^T \left[\boldsymbol{J}_{f_m}(\boldsymbol{x}_m), \boldsymbol{J}_{f_mb1}(\boldsymbol{x}_{b1}), \cdots \boldsymbol{J}_{f_mbL}(\boldsymbol{x}_{bL})\right] \quad , \quad (5)$$

for the $m^{\text{th}}$ GP expert. We include the neighboring GP experts $\boldsymbol{J}_{f_mb1}, ..., \boldsymbol{J}_{f_mbL}$ into the prior of the $m^{\text{th}}$ expert. This includes the information of neighboring GPs into the prior of the $m^{\text{th}}$ GP expert. Intuitively, the discontinuities occur at the shared boundary regimes of two GP experts. Our GP prior removes such boundary regimes locally by taking into account the neighboring regimes.

**Active uncertainty learning** The complexity of each GP expert is bounded to $O(N^3_{patch,m})$, where the number of data points $N_{patch,m}$ includes the data points of neighboring GPs: $N_{patch,m} = N_m + N_{b1} + \cdots + N_{bk}$, resulting in added costs $N_m < N_{patch,m}$. To mitigate, we perform active learning on the neighboring GP experts, and thus choose fewer, but the most informative points, e.g. IVMs [54, 55]. Intuitively, as the neighboring data are only included for reducing the discontinuity problem, we may select fewer data points. Concretely, we use the following steps: (i) for the neighboring GPs of the $m^{\text{th}}$ expert, we draw an initial subset and train the GPs. (ii) Using the trained GP models, we rank the remaining data (stored as a pool) by uncertainty (queries generated by uncertainty sampling). (iii) the GP expert is then updated with the most uncertain point. These steps repeat until $N_{bk}$ is reduced to a desired, smaller value. In this way, while forming a patchwork prior, the neighboring GPs actively choose the neighboring data they want to learn from. As a result, we obtain: $\bar{N}_{patch,m} < N_{patch,m}$.

**Gaussian process compression** We further reduce the complexity of the algorithm by exploiting model compression techniques. Note that the Jacobians $\boldsymbol{J}_f(\boldsymbol{x})$ are $N_{patch,m}$ by $P$ matrices where $P$ is the total number of parameters in DNNs. As $\boldsymbol{J}_f(\boldsymbol{x})$ represents the sensitivity of each parameter to the output, the elements of $\boldsymbol{J}_f(\boldsymbol{x})$ that belong to an unimportant DNN parameter can be removed. To do so, we propose a two-staged compression technique: (i) rank the DNN parameters by their importance using existing pruning methods and remove the corresponding elements of $\boldsymbol{J}_f(\boldsymbol{x})$ for all $m$. (ii) Per GP expert, rank the elements of $\boldsymbol{J}_{f_m}(\boldsymbol{x})$ by a metric $|\sum \boldsymbol{J}_{f_m}(\boldsymbol{x})|$, as smaller absolute values contribute less to the kernel. The first step is targeted at removing redundant parameters in DNNs (for the division step) while the second step is targeted at each individual expert, resulting in $\bar{P} < P$.

## 4 Experiments and Evaluations

We provide 5 sets of experiments to not only validate the method, but also to show the benefits of the proposed formulation namely performance, scalability and run-time. Implementation details can be found in the Appendix, and importantly, the accompanying video shows the experiments with a MAV.

**Toy illustration** With a toy regression on the Snelson dataset, we show (i) on how MoE-GPs can estimate the predictive uncertainty of DNNs, and (ii) the ability of the patchwork prior to produce smoother uncertainty estimates. For this, we train a single hidden layer Multi-layer perceptron

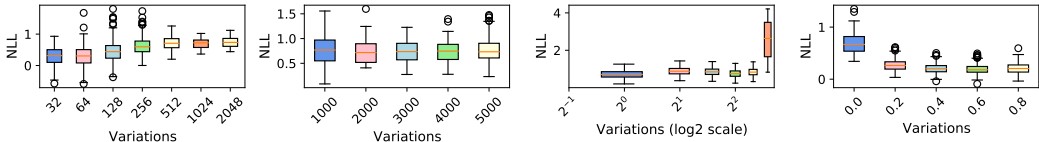

Figure 5: The effects of each hyperparameter is shown with normalized test NLL, by varying them in different steps (Variations), and fixing the others to default settings. Lower the better.

Table 1: The results of inverse dynamic experiments are reported using NLL. Lower the better.

| Train | Test | Ensemble | MC-dropout | rMC-dropout | Laplace | rLaplace | Ours |
|-------|------|----------|------------|-------------|---------|----------|------|
| sarcos | sarcos | *1.261±0.105* | 1.398±0.012 | 1.398±0.012 | 1.392±0.014 | 1.392±0.014 | **1.035±0.039** |
| sarcos | kuka1 | 17.86±2.995 | **16.21±0.866** | 20.58±1.144 | 25.88±1.354 | 24.55±2.746 | 21.53±2.268 |
| sarcos | kuka2 | 17.50±2.895 | **15.63±0.858** | 20.17±0.988 | 25.59±1.079 | 24.42±2.257 | 20.92±2.236 |
| sarcos | kukasim | **23.06±2.649** | 51.09±11.14 | 61.40±12.921 | 77.13±15.40 | 74.03±13.23 | 68.95±4.003 |
| kuka1 | kuka1 | 2.013±0.020 | *1.611±0.007* | 1.620±0.006 | 1.676±0.013 | 1.719±0.071 | **1.347±0.013** |
| kuka1 | kuka2 | 2.085±0.005 | 1.349±0.019 | 1.330±0.006 | **1.310±0.005** | **1.310±0.008** | *1.315±0.003* |
| kuka1 | kukasim | 60.44±1.108 | *42.14±2.869* | 48.76±0.566 | 60.37±0.799 | 59.74±1.571 | **1.348±0.012** |
| kuka1 | sarcos | *8.128±0.169* | 22.24±4.288 | 42.36±2.398 | 122.68±13.32 | 1837±2431.3 | **1.356±0.014** |
| kuka2 | kuka2 | 2.042±0.009 | 1.443±0.005 | *1.423±0.006* | 1.429±0.006 | *1.423±0.006* | **1.354±0.013** |
| kuka2 | kukasim | 63.07±0.741 | *38.94±0.561* | 48.93±0.779 | 60.36±1.005 | 59.05±1.158 | **1.333±0.012** |
| kuka2 | sarcos | *8.509±0.461* | 18.24±1.599 | 53.24±6.287 | 141.7±17.14 | 211.3±180.2 | **1.355±0.014** |
| kuka2 | kuka1 | 2.106±0.010 | 1.461±0.004 | 1.395±0.004 | *1.373±0.005* | *1.374±0.004* | **1.331±0.013** |

(MLP) with 200 units and a tanh activation. Evaluating for out-of-distribution (OOD) data (points far from train data), and domain-shift (DS) data (removed in-between points), as shown in figure 4, the produced uncertainty estimates are high as test points move away from the train data. Moreover, when DNN predictions match the train data, the produced uncertainty estimates are close to the data noise. We also observe that the patchwork prior mitigates the discontinuity problem of local GPs.

**On hyperparameters** Next, we examine (i) the influences of hyperparameter choices, and (ii) provide a simple tuning recipe. For this, we examine a regression task using a 5-layered, 200 units MLP on SARCOS robot arm data-set [38]. Here, we vary each hyperparameter while fixing the remaining ones, and we compute the Negative Log Likelihood (NLL). The results are shown in figure 5, with 4 hyperparmeters namely, (i) the number of GP experts, (ii) the subset sizes for clustering, (iii) the pruning level, and (v) the size of the informative data points. We observe the following: (i) the NLL is proportional to the number of GP experts, (ii) the subset size do not significantly influence the NLL, (iii) pruning steps have a tipping point where the NLL increases, and (iv) the active learning parameter has also a tipping point, where the NLL decrease is marginal. In contrary, by the design, the computational complexity increases with (i) lower number of experts and (ii) the size of actively selected points, and decreases with the pruning steps. We find these results confirm our intuition. For example, keeping 2048 GP experts for SARCOS results in 21 data-points per experts on average. This setting is inferior to keeping 32 GP experts, which has 1390 data points on average. Reflecting this, our strategy is to assign more data points for each GP expert, while selecting only the needed active learning points. The pruning levels can then be varied within a range that does not deteriorate the performance in order to reduce the computational complexity of the method.

**Comparison study** Now, we evaluate the performance of our method using the datasets namely SARCOS, KUKA1 and KUKA2 [42]. The baselines are MC-dropout [3], Laplace Approximation [47, 49], and their fast variants: rMC-dropout [28] and rLaplace [27, 29, 49]. The fast variants are our main competitors: principled Bayesian approaches that are training-free, i.e. works with pre-trained DNNs without modifications to the training procedures. We find decoupling DNN training to uncertainty estimates crucial, as it ensures comparisons of uncertainty estimates only by keeping the accuracy of the predictors constant amongst the baselines (in this case, the same pretrained 5-layered MLP across the baselines). We also include the ensembles [4]. Lastly, we adopt zero-shot cross-dataset transfer to systematically evaluate in-domain, OOD and DS scenarios, following the recent insights, that is, we train on a dataset, and evaluate uncertainty estimates for all other datasets.

The results are in table 1, where we averaged over 3 random initializations and report NLL as a metric (a standard for regression tasks). We find that our method often outperforms other baselines

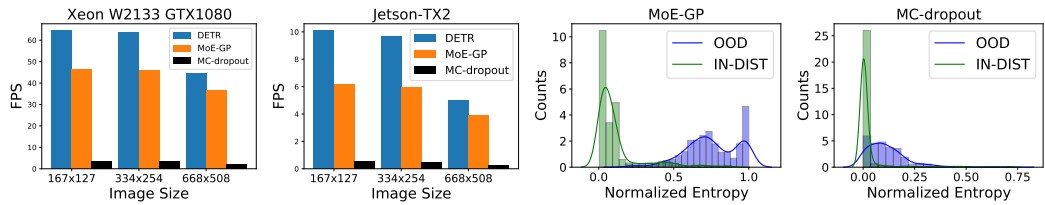

Figure 6: The run-time analysis (left) shows efficiency of our approach when compared to MC-dropout [3] with 20 samples. Any difference to a deterministic DNN can also be seen. The entropy histograms (right) show a novelty detection performance. The memory requirement is 8GB.

significantly. MC-dropout dominates in 3 experiments. Yet, MC-dropout requires sampling for uncertainty estimates - one of the core problems addressed in our work. We note again, that all the compared methods are training-free, making the comparisons meaningful. Importantly, rLaplace uses the NLM from section 3 to infer Gaussian uncertainty in weight space, in contrast to function space inference with GPs. Thus, the comparison of rLaplace and ours instantiates the comparison of inference between weight space and function space views. We believe that our results show the relevance of GP based uncertainty estimates for DNNs over uncertainty propagation methods.

**On scalability** Here, we show that (i) our approach scales to KUKA SIM dataset (contains 1984950 data points), and (ii) supports distributed training. Scalability is relevant, as DNNs operate in the regime of big data, and an exact GP does not scale to such settings. Moreover, a key benefit of MoE-GP over other sparse GPs lies in its distributed, local nature, and we can exploit it in practice. The results are depicted in figure 4, where we plot the training time in hours against the number of used GPUs. Using 512 GP experts and 100 iterations for MLL optimization, we find that our method scales to approximately 2 million data points within a day, and less than 8 hours when using 6 GPUs. This experiment shows an empirical evidence that our method scales to 2 million data points for the task of inverse dynamic learning, and validates the claim that MoE-GPs support distributed training.

**On a real robot** To validate the applicability of our method to a robot and evaluate the run-time on the hardware, we perform experiments on a MAV within the context of an on-going space demo mission for future planetary exploration [56]. For this, we train an object detector, DETR [57] with EfficientNet backbone. For testing, we create two scenarios: (i) a carry test, used for training and testing with 1008 manual labels, and (ii) flight tests with different configurations of the objects (e.g. space rover and lander). The later creates OOD samples, with a slight domain shift in data distribution. In field robotics, the assumption that the test set comes from the same training data distribution is routinely violated due to the changes in the environments, and we attempt to create such effects. Moreover, we also evaluate the complexity of our method on a Jetson-TX2, which is used on our MAV. The quantitative results are shown in figure 6, which shows that (i) our method can be fast on a Jetson-TX2, and (ii) can clearly separate OOD samples within this scenario. The accompanying video explains this setup and the qualitative results. More details can be found in the Appendix.

## 5 Discussion

If the computational complexity of GPs can be tamed in practice, our work shows that the predictive uncertainty of DNNs can be obtained from the GP formulation. By advancing the scalability of a full NTK, the advantages of our approach are demonstrated. Yet, the applicability to larger data regimes, e.g. imagenet, is confined to that of the sparse GPs, e.g. the non-parametric models require an access to the training data at run-time, which can require more memory than the parametric models such as DNNs. Sparse GPs may also face struggles when the output dimension is larger. The alternatives can be the weight space-based methods [58] such as Sharma et al. [59]. Here, various approximations have been devised so far, in order to cope with the high dimensional weight space. For robotics though, when the availability of data is limited, the dimensions of data space can be smaller than the weight space, and thus, our work can provide a reference that the Bayesian non-parametric can be a relevant tool in addressing the current challenges of the uncertainty estimation for neural networks.

In future works, we plan to apply the proposed methodology in a field robotics setting, and study how the uncertainty estimates can be used to improve the robustness of the robotic systems.

**Acknowledgments**

We thank the anonymous reviewers and area chairs for their thoughtful feedback. Special thanks to Klaus Strobl, Maximilian Denninger and Antonin Raffin for proof reading the paper, and also Seok Joon Kim for the support during his internship at DLR. We also would like to acknowledge the support of Helmholtz Association, the project ARCHES (contract number ZT-0033), the Initiative and Networking Fund (INF) under the Helmholtz AI platform grant agreement (ID ZT-I-PF-5-1) and lastly, the EU Horizon 2020 project RIMA under the grant agreement number 824990. Jianxiang Feng is supported by the Munich School for Data Science (MUDS) and Rudolph Triebel is a member of MUDS.

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
