# OpenReview forum: "Trust Your Robots! Predictive Uncertainty Estimation of Neural Networks with Sparse Gaussian Processes"
_robot-learning.org/CoRL/2021/Conference — CoRL2021 Poster_

### Official Review · Reviewer_Q7n6 · 2021-07-16

**Originality:** Good
**Technical Quality:** Good
**Clarity Of Presentation:** Poor
**Impact:** 3

**Recommendation:**

Weak Reject: I recommend rejecting the paper, but will not argue for my recommendation if the majority of other reviewers have a different opinion.

**Summary:**

The paper is motivated by predictive uncertainty that can be used on mobile robotic platforms (in this case MAVs).
The proposed model in a mixture-of-experts of BNNs, specfically the neural tangent kernel.
The model is evaluted on inverse dynamics learning and object detection.

**Issues:**

See above

**Reviewer Expertise:**

Very good: Comprehensive knowledge of the area

**Strengths And Weaknesses:**

Fast predictive models with useful uncertainty quantification would be a useful tool in robotics.
Improving the uncertainty quantification of BNNs to be as good as GPs is an active topic of research.

My central issue with this paper is that I couldn't tell if it was a BNN paper or robotics paper.
The real robot experiment and motivation is cool and relevant to CoRL, while the actual contribution of the paper is as a BNN.
Moreover, w.r.t. to the robotic applications, it is not clear what uncertainty is needed for, i.e. what is the downstream task (active learning, Bayesian optimization, etc)?
As a BNN paper, the baselines and experiments were somewhat limited compared to standard studies.

1) The neural linear model is actually an established class of Bayesian neural network (i.e. [1,2]), so you need a different term. This family should be included in the related work.

2) As I understand it, the NTK is more a tool for understanding the NN/GP connection rather than a practical model, since it comes with the kernel scaling issues and the need to compute Jacobians.

3) The core issue with this paper is that key design decsisions are centered on making BNNs like GPs, but GPs are never evaluated as a baseline. Since so much work is done to make the NTK scale to the small partioned small data, why aren't GP's evaluated for regression? The paper does not cite [3], which is a GP version of the idea presented here.

4) I was frequently confused by the memory requirements of your problem setting. Ensembles and GPs are described as being too memory intensive, and yet the MoE-GP model seems to cache an awful lot of memory (pre-computing various quanities and lots of models). I would like to see a comparison between the memory footprint of a MoE-GP vs ensembles and GPs to validate these claims.

5). 'To our knowledge, we are the first to show that you can use GPs to estimate the uncertainty of DNNs on a Jetson TX2 of a MAV.' While this is probably true, this is not a contribution worth stating, as it is too specific to be noteworthy.



[1] Benchmarking the Neural Linear Model, Ober et al AABI

[2] Neural Linear Models with Gaussian Process Priors, Watson et al, AABI

[3] Incremental Local Gaussian Regression, Meier et. al NeurIPS 2014


Minor

'eigen-vector' should just be eigenvector

typo on theory in appedix line 194

**Summary Of Recommendation:**

This paper has lots of interesting ideas, but perhaps too many as the current draft does not carefully motivate and validate the many design choices that go into the model.

I would be happy to increase the score if these design choices become more coherent.

---

> ### Author Response · Authors · 2021-08-26
> **Response to Reviewer Q7n6 (1/6)**
>
>
> We thank the reviewer for the valuable time and feedback. Our line-by-line responses to the reviewer are provided below.
>
> - **_On “My central issue with this paper is that I coudnt tell if it was a BNN paper or robotics paper. The real robot experiment and motivation is cool and relevant to CoRL, while the actual contribution of the paper is as a BNN”_**
>
> We thank the reviewer for the valuable opinion.
>
> Our research is at the intersection of robotics and machine learning (ML). As articulated in our related work (section 2), the primary focus of the paper is on robotic introspection [1], i.e. a cognitive ability of a robot to reason about “when they dont know” in order to avoid catastrophic failures. One of the approaches for robotic introspection are based on uncertainty estimation, where the learned models such as neural networks use probability theory to express their own reliability. Thus, this line of robotics research inherently lies at the intersection of robotics and ML –  leading to our contribution as a Bayesian method for robotic introspection.
>
> We would appreciate if the reviewer could enlighten us further on this comment.
>
> [1] Introspective Classification for Robot Perception, Grimmett et al, IJRR, 2015.
>
> - **_On “Moreover, w.r.t to the robotic applications, it is not clear what uncertainty is needed for, i.e. what is the downstream task (active learning, Bayesian optimization, etc)?”_**
>
> In our research, uncertainty is needed for robotic introspection as articulated in our related work (section 2) and thus, we do not consider a specific task such as active learning. Instead, we are interested in realizing an introspective capability of a robot, and we aim to understand how to engineer the given cognitive ability within a physical system. This requires both a systems research as well as a fundamental methodological research for robotics, as oppose to applying a general purpose ML technique to a robotic task or benchmark. Thus, our paper provides a long stretch from theory, algorithm to deployments on a real robot, and serve as a reference on what is possible, and what might be the limiting factor for further algorithmic developments. With this, we hope to go beyond robotics as a downstream application of ML, but for the development of BNNs that are more applicable to robotic introspection.
>
> We would appreciate if the reviewer could enlighten us further on this comment.
>
> - **_On “As a BNN paper, the baselines and experiments were somewhat limited compared to standard studies.”_**
>
> We thank the reviewer for the valuable opinion.
>
> As a paper on a Bayesian method for robotic introspection, we think that our experiment designs are relevant. From our experience, many studies on Bayesian Neural Networks (BNNs) often use UCI, MNIST or CIFAR10 for the experiment setup. We refer to table 5 of the recent survey [1] as an example. In this regard, a Bayesian method that is validated on a robot can be of interest to the CORL audience as these benchmarks such as UCI may not close the gap in applicability of these methods to a real robot. Moreover, our paper contains 3 ablation studies, 1 comparison study to 5 other methods across 3 datasets, and a real robot experiment. So, we hope the reviewer appreciates the extensiveness of our study as a paper on a Bayesian method for robotics.
>
> We would appreciate if the reviewer could enlighten us further on this comment.
>
> [1] A Survey of Uncertainty in Deep Neural Networks, Gawlikowski et al, 2021.
>
> - **_On “1. The neural linear model is actually an established class of Bayesian Neural Networks, so you need a different term. This family should be included in the related work.”_**
>
> To clarify, the neural linear model in the given references [1,2] are the same class of model as we introduce in section 3. As reference [2] states, a neural linear model is “a Bayesian linear regression in a projected feature space, where the projection is learned by a neural network”.  In our case, the Jacobian is the projection that is learned by a neural network. While the references [1,2] sometimes refer to a neural linear model as using only the last layer, earlier papers [3-5] assume the first order Taylor series expansion around the last layer, not necessarily using only the last layer.
>
> We revised the corresponding part in section 3 and added the suggested references (lines 95-109). We are also open to using more generic terms such as the generalized linear model (GLM).
>
> [1] Benchmarking the Neural Linear Model, Ober et al, AABI 2020.
>
> [2] Neural Linear Models with Gaussian Process Priors, Watson et al, AABI 2020.
>
> [3] Bayesian Approach to Neural-Network Modeling with Input Uncertainty, Wright, IEEE transactions on neural networks 1999.
>
> [4] Information-based objective functions for active data selection, MacKay, Neural Computation  1992.
>
> [5] (Chapter 5.7) Pattern recognition and machine learning, Bishop, Springer 2006.

---

> > ### Author Response · Authors · 2021-08-26
> > **Response to Reviewer Q7n6 (2/6)**
> >
> >
> > - **_On “2. As I understand it, the NTK is more a tool for understanding the NN/GP connection rather than a practical model, since it comes with the kernel scaling issues and the need to compute Jacobians.”_**
> >
> > We thank the reviewer for the valuable opinion. However, we disagree that the NTK is limited as a tool for theoretic work.
> >
> > There has been encouraging recent results in applying the Neural Tangent Kernel (NTK) for uncertainty estimation [1,2] (as discussed in section 2, lines 59-63) and in this context of on-going research on the NTK, our work highlights the applicability of the NTK for the real-time uncertainty estimation in robotics. As there has been advances in kernel approximation methods, it can be a relevant question to find – to what extent we can engineer the NTK in practical settings of robotics, in order to obtain its benefits. For the results, we show the possible areas of its application (e.g. object detection on a MAV). This is achieved by addressing the limitations of the kernel scaling issues via the mixtures of experts idea, model compression and active learning. We also highlight that the Jacobian computations are needed for many non-NTK approaches for sampling-free uncertainty estimation of neural networks [3, 4] and more efficient computation routines are being currently developed by the community. In this regard, we find the new software such as JAX [5] to a promising solution in the near future.
> >
> > Specifically to the kernel scaling issues, we think that the NTK can be more suited for many use-cases in robotics, that are characterized by the limited amount of annotated data. This is in comparison to the weight-space formulation of Bayesian Neural Networks (BNNs). To explain our stance, consider an object detection scenario, where a developer has to create the labeled images for transfer learning. The NTK needs to then deal with the kernel scaling issues, which is associated with the number of data points, say 10000 images. An alternative is to deal with the scaling issues of BNNs in weight space, where the dimensions of network parameters are more than 1 million. Both approaches require various approximations to be applicable, but in many domains of robotics - due to the limited amount of available data - dealing with the kernel scaling issue can be more practical than dealing with the approximations in the weight space (e.g. Bernoulli posterior in MC-dropout). In this regard, our experiments provide the relevance of this function-space view for robotics (figure 6). By dealing with the challenges of the NTK to an extent, our real robot experiments show that the NTK can perform better than MC-dropout in terms of novelty detection and run-time.
> >
> > We would appreciate if the reviewer could enlighten us further on this comment.
> >
> > [1] Bayesian deep ensembles via the neural tangent kernel, He et al, NeurIPs 2020.
> >
> > [2] Exploring the uncertainty properties of neural network’s implicit priors in the infinite-width limit, Adlam et al, ICLR 2021.
> >
> > [3] Sampling-free epistemic uncertainty estimation using approximated variance propagation, Postels et al, ICCV 2019.
> >
> > [4] Benchmarking the Neural Linear Model, Ober et al, AABI 2020.
> >
> > [5] JAX: https://github.com/google/jax

---

> > > ### Author Response · Authors · 2021-08-26
> > > **Response to Reviewer Q7n6 (3/6)**
> > >
> > >
> > > - **_On “3. The core issue with this paper is that key design decisions are centered on making BNNs like GPs, but GPs are never evaluated as a baseline. Since so much work is done to make the NTK scale to the small partioned small data, why aren't GP's evaluated for regression?”_**
> > >
> > > The reason for not evaluating a full GP as a baseline in the comparative study (table 1) is scalability. To explain, standard GPs do not scale to the inverse dynamics tasks, where the number of data points range from approximately 40000 to 120000. To back up this claim as a norm in GP literature, we point to the references [1-3]. Specifically, the survey of Liu et al [1] discusses the computational intractability of GPs and summarizes the techniques of scalable GPs including their computational complexity. Table 2 of Liu et al [1] assumes more than 10000 as “big regression datasets” where a standard GP does not scale. Similarly, in chapter 8 of Rasmussen et al [2], the textbook states that for large problems with more than 10000 data points as an example, the computational complexity of a full GP is prohibitive. Lastly, in the paper referred by the reviewer – Meier et al [3], the authors also use the inverse dynamic settings namely sarcos and kuka, but do not compare to a full GP (instead choosing an online learning with sparse approximations [4]). To our knowledge, there exists no implementation of a full GP (without approximations) in the inverse dynamics task we consider in our paper.
> > >
> > > Nevertheless, as a validation of the concept, we provide a full GP with the NTK and the RBF as a baseline, which can be found in figure 2-3 of the appendix. Our experiments are on toy regression where a standard GP can scale in terms of computational tractability. The results show that our approach can closely approximate the full GP with the NTK in this setting. We note that we are trying to obtain a model with the neural network predictions, and estimate its uncertainty with the mixtures of GP experts with the NTK (as illustrated in figure 1). This validation experiment shows that the uncertainty estimates of mixtures of GP experts with the NTK can obtain a similar profile of uncertainty estimates to that of a full GP with the NTK. To be more precise, as shown in our ablation studies, an excessive partitioning of the data tends to the deviation of uncertainty estimates. Thus, in practical settings, we have provided the comparison studies to other existing uncertainty quantification techniques for neural networks, in order to investigate if our model can still be useful when compared to the given baselines.
> > >
> > > We thank the reviewer for a valuable comment, and we are open for further discussion.
> > >
> > > [1] When Gaussian Process Meet Big Data: A Review of Scalable GPs, Liu et al, IEEE transactions on neural networks and learning systems, 2020.
> > >
> > > [2] (Chapter 8) Gaussian Processes for Machine Learning, Rasmussen et al. MIT Press 2006.
> > >
> > > [3] Incremental Local Gaussian Regression, Meier et al, NeurIPs 2014.
> > >
> > > [4] Real-time model learning using incremental sparse spec-turn Gaussian process regression. Gijsberts et al, Neural Networks 2013.

---

> > > > ### Author Response · Authors · 2021-08-26
> > > > **Response to Reviewer Q7n6 (4/6)**
> > > >
> > > >
> > > > - **_On “The paper does not cite [3], which is a GP version of the idea presented here.”_**
> > > >
> > > > Maier et al [1] is not a GP version of the idea presented here. To clarify the confusion, both Maier et al [1] and our approach belong to the same class of scalable GPs called mixtures of GP experts or local GPs, which is one of the well established models in the GP literature [2]. Hence, we have instead cited the early papers that originally introduced the idea, namely Jacobs et al [3] and Tresp [4], and discussed other papers that we have closely build upon, namely Park et al [5-6] (as described in our related work). To avoid any confusion, we have also included the discussions on Maier et al [3] in the revised manuscript (lines 129-130).
> > > >
> > > > To clarify the similarity and the differences, Maier et al [1] is similar to ours in the sense that the data is partitioned into smaller subsets where individual smaller GPs are trained (the idea of mixtures of experts model [3]), in order to improve the computational scalability of a full GP. The differences are in the training step and the testing step. Maier et al [1] proposes variational inference for partitioning the data and training the GP experts (section 3 of [1]) whereas we use a clustering technique and the marginal likelihood optimization (similar to [5-6]). For the predictions, Maier et al [1] utilize Bayesian model averaging (section 3.1.3 of [1]), while we use a hard partitioned aggregation step (similar to [5-6]).
> > > >
> > > > We thank the reviewer for a valuable comment, and we are open for further discussion.
> > > >
> > > > [1] Incremental Local Gaussian Regression, Meier et al, NeurIPs 2014.
> > > >
> > > > [2] When Gaussian Process Meet Big Data: A Review of Scalable GPs, Liu et al, IEEE transactions on neural networks and learning systems, 2020.
> > > >
> > > > [3] Adaptive mixtures of local experts, Jacobs et al, Neural Computations 1991.
> > > >
> > > > [4] Mixtures of Gaussian processes, Tresp, NeurIPs 2001.
> > > >
> > > > [5] Efficient Computation of Gaussian Process Regression for Large Spatial Data Sets by Patching Local Gaussian Processes, Park et al, JMLR 2016.
> > > >
> > > > [6] Patchwork kridging for large-scale Gaussian process regression, Park et al, JMLR 2018.

---

> > > > > ### Author Response · Authors · 2021-08-26
> > > > > **Response to Reviewer Q7n6 (5/6)**
> > > > >
> > > > >
> > > > >
> > > > > - **_On “4. I was frequently confused by the memory requirements of your problem setting. Ensembles and GPs are described as being too memory intensive, and yet the MoE-GP model seems to cache an awful lot of memory (pre-computing various quanities and lots of models). I would like to see a comparison between the memory footprint of a MoE-GP vs ensembles and GPs to validate these claims.”_**
> > > > >
> > > > > The memory requirements of our problem setting is as follows. A Jetson TX2 [1] (the hardware on our robot) has 8GB of GPU memory at run-time. This is a strict memory requirement of our problem setting while the benchmarks using desktops and GPU servers usually do not pose such strict requirements. In light of these, we provide the relevant information below for the real robot experiments.
> > > > >
> > > > > Deep Ensembles:
> > > > >
> > > > > For the validation of the claim, our object detector uses 5.61 GB at run-time. As the memory of deep ensemble scales linearly with the ensemble size, using an ensemble of 5 requires 28.05 GB, and an ensemble of 15 require 84.15 GB. Therefore, in our case, we cannot store more than 1 ensemble member simultaneously on a GPU.
> > > > >
> > > > > For the comparison, our approach does not outperform deep ensembles in terms of the memory in all the use-cases. To explain, the memory consumption of our approach heavily depends on the amount of available data, while deep ensemble depends on the size of neural networks. Thus, we acknowledge that there are use-cases where deep ensembles can outperform our method in terms of the memory, e.g, when the memory due to the size of data exceeds the memory due to the size of neural networks.
> > > > >
> > > > > GPs:
> > > > >
> > > > > For the validation of the claim, we stated that “the computational costs of full GPs are intractable O(N^3)”, where N is the number of training data point [4, 5]. To explain, a GP requires the run-time storage and the run-time inversion of the kernel matrix, which involves the memory of O(N^2) and O(N^3) respectively [4, 5]. Therefore, if we have 50000 data points, the memory required for a GP is equivalent to storing and inverting a matrix of 50000 by 50000 at run-time. As a result, the storage alone costs 20GB of memory assuming double precision, and the inversion will cost more. Thus, the computational costs of full GPs are known to be intractable and we further refer to the point 3 of this review.
> > > > >
> > > > > MoE-GPs:
> > > > >
> > > > > The memory required for MoE-GPs with a hard gating function are O(K*M^2) for the kernel matrix storage and O(M^3) for the kernel matrix inversion, where K is the number of experts, and M is the number of data points for each GP experts. Intuitively, at run-time, we store K matrices of size MxM, and invert a MxM matrix due to the hard partitioned aggregation. Similar to before, say we have 50000 data points. Forming 10 GP experts, assume we have K=10 and M=5000. The full storage is then  2GB instead of 20GB, and we only need the inversion of 5000 x 5000 matrix. We can also increase K to a higher value, then we need the storage and the inversion of many smaller matrices, which is cheaper than operating on a larger matrix. The same principle applies to the Jacobian, i.e. apart from pruning and active selection, having to store smaller matrices due to the division of data is beneficial as the matrix storage costs quadratic to the number of matrix elements.
> > > > >
> > > > > In terms of the specific values, figure 4 of the appendix shows the memory analysis of our approach on the Jetson TX2 at run-time. The dynamic computation graph is utilized here, and the Jacobian of the entire neural network is used.
> > > > >
> > > > > We have revised our manuscript to address this point.
> > > > >
> > > > > [1] https://developer.nvidia.com/embedded/jetson-tx2
> > > > >
> > > > > [2] A Survey of Uncertainty in Deep Neural Networks, Gawlikowski et al, 2021.
> > > > >
> > > > > [3] A Review and Comparative Study on Probabilistic Object Detection in Autonomous Driving, Feng et al, 2021.
> > > > >
> > > > > [4] When Gaussian Process Meet Big Data: A Review of Scalable GPs, Liu et al, IEEE transactions on neural networks and learning systems, 2020.
> > > > >
> > > > > [5] (Chapter 8) Gaussian Processes for Machine Learning, Rasmussen et al. MIT Press 2006.
> > > > >
> > > > >
> > > > > - **_On “5. To our knowledge, we are the first to show that you can use GPs to estimate the uncertainty of DNNs on a Jetson TX2 of a MAV.' While this is probably true, this is not a contribution worth stating, as it is too specific to be noteworthy.”_**
> > > > >
> > > > > We thank the reviewer for the valuable opinion. Our list of contributions are revised accordingly.

---

> > > > > > ### Author Response · Authors · 2021-08-26
> > > > > > **Response to Reviewer Q7n6 (6/6)**
> > > > > >
> > > > > >
> > > > > > - **_On “eigen-vector' should just be eigenvector”_**
> > > > > >
> > > > > > We thank the reviewer for pointing out. We have incorporated this comment in the revised manuscript.
> > > > > >
> > > > > > - **_On “typo on theory in appedix line 194”_**
> > > > > >
> > > > > > We thank the reviewer for pointing out. We have incorporated this comment in the revised manuscript.
> > > > > >
> > > > > > - **_On “This paper has lots of interesting ideas, but perhaps too many as the current draft does not carefully motivate and validate the many design choices that go into the model.”_**
> > > > > >
> > > > > > In the revised manuscript, we have highlighted (in green) the sentences that motivate and validate the design choices that go into the model. The comments on validation against full GPs are addressed in our response to the point 3 above. If the reviewer has more specific concern on a particular design choice, we can provide more discussions and data during the rebuttal period and beyond.

---

> > > > > > ### Comment · Reviewer_Q7n6 · 2021-09-01
> > > > > > **Response 5/6**
> > > > > >
> > > > > > Thanks for including this memory footprint discussion, I believe it improves the paper.

---

> > > > > ### Comment · Reviewer_Q7n6 · 2021-09-01
> > > > > **Response 4/6**
> > > > >
> > > > > Thank you for including a discussion of Meier et al's work. While I appreciate there are some implementation difference, I hope you understand that by 'GP version of the idea presented here' I meant MoE + GP rather than exactly the same. Moreoever, while they may not have had the original idea, I think it's still related work worth discussing.

---

> > > > ### Comment · Reviewer_Q7n6 · 2021-09-01
> > > > **Response 3/6**
> > > >
> > > > Thanks for this additional experiment, I think it greatly improves the paper

---

> > > ### Comment · Reviewer_Q7n6 · 2021-09-01
> > > **Response 2/6**
> > >
> > > I can concede that perhaps the NTK can be practical with this MoE approach to aid scaling. Again, performing the standard BNN benchmarks would really help confirm performance.

---

> > ### Comment · Reviewer_Q7n6 · 2021-09-01
> > **Response (1/6)**
> >
> > Re: central issue and uncertainty
> > I fully appreciate the need of uncertainty quantification (UQ) for robotics. My question is who the paper is for, as the paper is somewhat awkwardly positioned as a BNN variant for applications in robotics.
> >
> > I appreciate the main motivation is for fast and low memory footprint for embedded applications. I think the authors have done a good job at this. What remains is
> > 1) how does this BNN compare to other BNN variants? UCI is useful for this. The datasets aren't that insightful (although there is one robot kinematics dataset) but they are well established in assessing BNN / GP performance.
> > 2) is the uncertainty from this model actually useful? Approximations in BNNs usually have downstream affects on the quality of the UQ. While metrics like log likelihood are useful they do not tell a clear picture. Therefore, to really assess UQ you need some kind of downstream task that uses the UQ. This does not mean you are necessarily interested in these tasks, but these tasks are an objective way of comparing UQ between models. Active learning, Bayesian optimization, model-based RL (e.g. PILCO), outlier rejection etc all require good OOD UQ in order to carry out the task effectively, therefore it is a good way of quantifying the quality of UQ between models.
> > Table 1 does not really convince me that the UQ is necessarily good, as where your model performs best (kuka*), the OOD test LLH is roughly the same as the in-distribution LLH, which suggests that it is potentially overconfident under distribution shift. This is one way LLH metrics can be gamed. For tasks like active learning and Bayesian optimization, this OOD overconfidence would directly result in poor performance.
> >
> > For the novelty histograms, these look good, but there needs to be a outlier rejection task (see A Systematic Comparison of Bayesian Deep Learning Robustness in Diabetic Retinopathy Tasks, Filos et al for an example) to properly assess this in a systematic way.
> >
> > Re: NLM. Linearizing the network is also a linear model but I would say it is separate from a NLM, as the features of the NLM should be outputs of a NN and optimized using type-II ML. See Improving predictions of Bayesian neural nets via local linearization by Immer et. al for a recent discussion on various linearization-based approaches.

---

> ### Comment · Reviewer_Q7n6 · 2021-09-01
> **General Response**
>
> I wish to thank the authors for their work in the rebuttal, I think the paper has been improved a lot.
>
> While I think the motivation has been clarified, looking at the new draft, my central issue remains w.r.t. standard BNN experiments and downstream UQ tasks. Therefore I leave my score unchanged. My feeling is that the main text needs to be more concise to explain the idea, so that there is more space available for the experiments.  I agree with hvPu that the paper needs work in clarity (or conciseness) of presentation and benefit of this approach's UQ.

---

### Official Review · Reviewer_hvPu · 2021-07-24

**Originality:** Good
**Technical Quality:** Good
**Clarity Of Presentation:** Good
**Impact:** 2

**Recommendation:**

Weak Accept: I recommend accepting the paper, but will not argue for my recommendation if the majority of other reviewers have a different opinion.

**Summary:**

This paper presents an approach to equipping trained DNNs with a measure of epistemic uncertainty. Leveraging a connection between the local liberalization of a DNN and the resulting equivalent GP (using the neural tangent kernel), the authors propose to use this GP to characterize the uncertainty on test points. The approach approximates this equivalent GP using a mixture-of-experts GP model, wherein a hard gating function divides the input into disjoint sets, each of which correspond to a separate GP. These GPs have a kernel with the same features equal to the Jacobian of the DNN. The approach requires (1) performing approximate kernel k-means to design the gating function and split the training data into subsets; (2) fitting a NTK GP to each of the subsets, by (2a) taking an active learning approach to include only a limited number of informative datapoints in the gram matrix for each GP; and (2b) pruning features by removing elements of the Jacobian which have small values and thus are less relevant to the kernel. To ensure smoothness across the separate GPs, each GP also includes the data from some of its neighbors. At test time, one can compute an uncertainty estimate by constructing evaluating the gating function to determine the expert GP to to evaluate, and subsequently evaluating the kernel for that GP on the test input and the selected training data for that expert, and finally, using this information to compute the posterior covariance.

The authors provide theoretical motivation for their approach justifying why the MoE GP is a good approximation of the DNN uncertainty.

**Issues:**

- The authors should, if possible, refactor the theoretical results as suggested in the main review section, or provide more justification why the posterior predictive uncertainty of the MoE-DNN model is a useful analog to the posterior predictive uncertainty of a single GP.
- Address the mismatch between the NLM derivation and the GP expressions used in practice (specifically, the choice of observation noise)
- The main missing experiments are comparisons to Deep Ensembles.
- More details on the stopping criterion for the active learning and Jacobian pruning strategies should be given.
- The discussion should highlight the fact that this method requires access to training data during the inference procedure.

**Reviewer Expertise:**

Very good: Comprehensive knowledge of the area

**Strengths And Weaknesses:**

This paper presents an interesting approach exploiting a connection between local linearization of a DNN and GPs. The presented approach takes this theoretical idea, and presents an algorithm that yields tractable inference times at test time. While the method is complex, the paper included helpful figures which communicated the ideas well. I especially liked Figure 3 in this regard. I also appreciated the practical results, demonstrating that the approach can be deployed in hardware, and offers good OOD detection performance over using mc-dropout.

However, there are several areas for improvement.

First, I found the paper to be occasionally hard to follow, especially in the theoretical derivations. The jump from DNN setup to the NLM was abrupt, without explanation providing intuition for the particular choice of covariances. The construction of the practical model also felt disconnected from the theoretical derivation, specifically, while the NLM treats the output as having additive gaussian noise with covariance equal to the inverse Hessian of the loss (eq. 1), a quantity that is input dependent, the predictions are made assuming equal noise for each sample of \sigma_0. Furthermore, eq. 3, connecting the GP regression uncertainty to a classification task, lacked theoretical justification beyond an intuitive notion that it would have the right effect on the predictive uncertainty. This felt at odds with the fully general derivation of the NLM.

The theoretical results could be presented more clearly. While I appreciated the visual description of the theoretical results in Figure 2, I felt that the choice to start by viewing a single DNN as a mixture of expert DNNs with identical networks for each expert, and subsequently showing that each expert can be cast as a GP. In contrast the premise of the approach was to approximate the posterior predictive uncertainty of the full DNN, in which case it would have been clearer to consider the GP corresponding to the full DNN, and subsequently show how the MoE-GP approximates the full GP. Furthermore, the paper could be strengthened by including theoretical results bounding the errors made by using the various approximation techniques applied to make the approach practical: the active learning for data subsampling, and the feature pruning.

Second, the scalability of this approach is unclear. While the authors take several steps to make the approach more practical, including subsampling data and breaking the data into subsets, the kernel-GP approach taken requires storing significant amounts of data at test time (even if all of the data isn’t used in evaluation). In addition, the authors use separate GPs per output dimension, meaning the approach cannot capture correlations across output dimensions, or scale easily to higher-dimensional outputs (e.g. semantic segmentation tasks). Also, while the authors demonstrate the approach on a large dataset in simulation, the on-robot results use a much smaller dataset. Furthermore, the appendix suggests that the results in the paper also only consider the Jacobian of the last few layers of the network, rather than the whole DNN. This hinders the authors’ argument for the scalability and efficiency of the approach.

Finally, the authors should compare to more standard benchmarks, in addition to MC-dropout. Experiments comparing to Deep Ensembles are crucial, as they are a commonly used tool for DNN uncertainty, and can give good results even with a low number of ensemble members, allowing for efficient real-time application. Furthermore, there are recent works considering different Laplace approximations that are closely related to a weight-space view of the DNN GP, that are worth discussing if not comparing against [1,2].


[1] Madras et al., “Detecting Extrapolation with Local Ensembles,” ICLR 2019
[2] Sharma et al., “Sketching Curvature for Efficient Out-of-Distribution Detection for Deep Neural Networks,” UAI 2021


**Summary Of Recommendation:**

Overall, I think this paper presents some interesting, novel ideas, but the message is unclear, and the practical implementation of the idea has some shortcomings in its ability to scale. The paper would be improved with improved clarity in the presentation of the ideas and results, and a more thorough experimental comparisons highlighting the benefit of this approach over existing baselines for predictive uncertainty estimation.

---
Having read the revisions, I believe the authors have addressed my issues and I have increased my score.

---

> ### Author Response · Authors · 2021-08-26
> **Response to Reviewer hvPu (1/4)**
>
>
> We thank the reviewer for the valuable time and feedback. Our line-by-line responses to the reviewer are provided below.
>
> **On improving the technical exposition:**
>
> - **_“The jump from DNN setup to the NLM was abrupt, without explanation providing intuition for the particular choice of covariance.”_**
>
> In the revised manuscript, we have added more explanations for better clarify (lines 100-106, highlighted in blue).
>
> We thank the reviewer for the valuable feedback.
>
> - **_“The construction of the practical model also felt disconnected from the theoretical derivation, specifically, while the NLM treats the output as having additive gaussian noise with covariance equal to the inverse Hessian of the loss, a quantity that is input dependent, the predictions are made assuming equal noise for each sample of sigma_0”_**
>
> - **_AND “Address the mismatch between the NLM derivation and the GP expressions used in practice (specifically, the choice of observation noise)”_**
>
> There is no mismatch between the NLM derivation and the GP expression used in practice, specifically the modelling assumptions on observation noise. In contrary, the derivation stems from the existing and established weight space vs. function space view of a generalized linear model, where GPs model the observation noise from data via the marginal log-likelihood optimization [1]. Also, if one assumes the mean squared error as a loss function, for example, the Hessian of the loss is an input-independent constant term.
>
> We have revised the corresponding section by adding more explanations (lines 135-139 highlighted in blue) and would like to thank the reviewer for the valuable feedback.
>
> [1] (Chapter 2) Gaussian Processes for Machine Learning, Rasmussen et al. MIT Press 2006.
>
> - **_“Furthermore, eq.3, connecting the GP regression uncertainty to a classification task, lacked theoretical justification beyond an intuitive notion that it would have the right effect on the predictive uncertainty. This felt at odds with the fully general derivation of the NLM”_**
>
> To address this issue, we more actively refer to a prior work that provides theoretical justification of this step, and in the revised manuscript, we further provide more detailed motivation (lines 146-148, highlighted in blue).
>
> We thank the reviewer for the valuable feedback.
>
> - **_“While I appreciated the visual description of the theoretical results in Figure 2, I felt that the choice to start by viewing a single DNN as a mixture of expert DNNs with identical networks for each expert, and subsequently showing that each expert can be cast as a GP. In contrast the premise of the approach was to approximate the posterior predictive uncertainty of the full DNN, in which case it would have been clearer to consider the GP corresponding to the full DNN, and subsequently show how the MoE-GP approximates the full GP.”_**
>
> In the revised manuscript, we address this comment by providing the terminology for each parts, e.g. instead of “Proposition 2.2 & Lemma 2.2”, we write “approximation step”, and modifying the figure 2 in order to avoid misleading the reader. We also provide more information in the caption in order to avoid any confusion and revised the section on introduction. We are also open to refactoring figure 2 and their explanations completely, if the reviewer finds the revision misleading.
>
> We thank the reviewer for the valuable feedback.
>
> - **_“Furthermore, the paper could be strengthened by including theoretical results bounding the errors made by using the various approximation techniques applied to make the approach practical: the active learning for data subsampling, and the feature pruning.”_**
>
> We thank the reviewer for the valuable opinion.
>
> We think that this is beyond the scope of this paper. The paper contains a long stretch from theory, algorithm to real robot experiments (as reviewer Q7n6 points to "many" ideas). Furthermore, the theory part of the paper is to provide a foundation for our approach more generally, and we stress that the designed algorithm in section 3.3 is only one way to exploit this theory. Our hope is that the foundation provided is useful for future researchers in further improving the algorithm including theoretical understandings on the approximations made in practice.

---

> > ### Author Response · Authors · 2021-08-26
> > **Response to Reviewer hvPu (2/4)**
> >
> >
> > - **_“The authors should, if possible, refactor the theoretical results as suggested in the main review section, or provide more justification why the posterior predictive uncertainty of the MoE-DNN model is a useful analog to the posterior predictive uncertainty of a single GP.”_**
> >
> > First, we have revised the presentation of the theoretical results in the light of the review. For this point, we refer to the corresponding part of the response above.
> >
> > For more justification, we provide an empirical result. In figure 2 and 3 of the appendix, we provide new results comparing the posterior predictive uncertainty of a single GP with neural network-based kernel, and the posterior predictive uncertainty of their approximation (MoE-GP). In this setting, the results confirm that MoE-GP can resemble the posterior predictive uncertainty of a single GP. Thus, we believe this provides more justification on this comment. We also note that the reviewers' description of our approach has centered on our contributions, and the MoE-DNN is needed for the theoretic derivations, not for practice in this work.
> >
> > We thank the reviewer for the valuable feedback.
> >
> > - **_“More details on the stopping criterion for the active learning and Jacobian pruning strategies should be given.”_**
> >
> > In the revised manuscript, we provide more implementation details for the active learning and Jacobian pruning strategies in section 4.5.2 (lines 641-650) of the appendix.
> >
> > We thank the reviewer for pointing this out.
> >
> >
> > **On scalability:**
> >
> > - **_“Second, the scalability of this approach is unclear.”_**
> >
> > - **_AND “This hinders the authors’ argument for the scalability and efficiency of the approach.”_**
> >
> > We thank the reviewer for the valuable opinion and provide the general clarifications on the scalability as follows.
> >
> > In the light of the reviews, we feel that the term “scalable Gaussian Processes”[1] could have been misleading. In the revised manuscript, we replace the term “scalable Gaussian Processes”[1] to “sparse Gaussian Processes”[2] which are analogous terms (the former being the term more recently used, while the latter being used more in the past). Furthermore, we also clarified that in the GP community, a dataset with more than 10000 points has been referred to large-scale dataset because a full GP does not scale to this regime [1,2, 3]. As our paper lies at the intersection of neural networks and GPs, we apologize for the confusion we have caused in the original manuscript. The limitations of our approach on scalability have also been significantly moved from the appendix to the main text. We thank the reviewer for addressing the important points for improving our paper.
> >
> > To explain our stance more clearly though, we define scalability as a relative measure. For example, ‘method A is scalable’ might be a misleading claim while the statements such as ‘method A is more scalable than method B’ or ‘method A scales to C number of data points’ are the claims that may better serve the current scientific standards. In light of this, the paper claims that our approach “offers an improved scalability when compared to a full Gaussian Process (GP) with a neural network-based kernel” (line 36-37), and “show that our method scales to approximately 2 million data points for the task of learning inverse dynamics (line 40-41)”. Our experiments validate these claims (section 4), substantiating the contribution on improving the scalability of GPs with a neural network-based kernel for robotics, and further demonstrating the benefits over the existing methods.
> >
> > The revised manuscript contains the clarifications on this point, and we are open for further suggestions and discussions.
> >
> > [1] When Gaussian Process Meet Big Data: A Review of Scalable GPs, Liu et al, IEEE transactions on neural networks and learning systems, 2020.
> >
> > [2] A Unifying View of Sparse Approximate Gaussian Process Regression, Candela et al, JMLR 2005.
> >
> > [3] (Chapter 8) Gaussian Processes for Machine Learning, Rasmussen et al. MIT Press 2006.

---

> > > ### Author Response · Authors · 2021-08-26
> > > **Response to Reviewer hvPu (3/4)**
> > >
> > >
> > > - **_“While the authors take several steps to make the approach more practical, including subsampling data and breaking the data into subsets, the kernel-GP approach taken requires storing significance amounts of data at test time”_**
> > >
> > > - **_AND “the discussion should highlight the fact that this method requires access to training data during the inference procedure”_**
> > >
> > > In favor of more results, we have discussed this point as our limitations in the appendix of the first submission. We apologize for the confusion. Now we have included this discussion in the main text (lines 290-300) where we expect this point to be the most visible.
> > >
> > > We acknowledge that Bayesian non-parametric techniques such as GPs require an access to some form of training data at deployment. On the other hand, these techniques can provide reliable uncertainty estimates [1,2], i.e, by examining how close is the new test data to the training data (via kernel function as an example). We argue that for the application scenarios where it is feasible to store the training data, we can obtain the benefits of Bayesian non-parametric techniques for neural networks. Therefore, our paper contributes to expanding the applicability of these techniques for robotics.
> > >
> > > We are open for further suggestions and discussions.
> > >
> > > [1] (Chapter 1) Gaussian Processes for Machine Learning, Rasmussen et al. MIT Press 2006.
> > >
> > > [2] Introspective Classification for Robot Perception, Hugo et al, IJRR, 2015.
> > >
> > > - **_“In additional, the authors use separate GPs per output dimension, meaning the approach cannot capture correlations across output dimensions, or scale easily to higher dimensional outputs”_**
> > >
> > > In the original manuscript, we have outlined this point as our limitations and provided future directions – to incorporate the state-of-the-art frameworks of multi-output GPs. In the revision, we have moved the discussion on this point to the main text and revised our limitations section in the appendix.
> > >
> > > We note however, that our approach still can outperform existing methods without capturing the correlations across output dimensions. Moreover, many important applications in robotics utilize low dimensional outputs, e.g. object pose estimation, visual odometry, etc. We argue that for the researchers working in these domains, our results can be relevant.
> > >
> > > We are open for further suggestions and discussions.
> > >
> > > - **_“Also, while the authors demonstrate the approach on a large dataset in simulation, the on-robot results use a much smaller datasets”_**
> > >
> > > The purpose of our real robot experiment is to validate if our approach can work on a physical robot, not to test the scalability. To explain, we had to perform the flight experiments to collect the data, and manually label the images for transfer learning. We stopped creating more labeled data once we could realize the demonstration (seen in the attached video). As creating the relatively larger dataset is both infeasible and not necessary in our scenario, we decided to test the scalability of our approach in other experiments, e.g. using kuka sim dataset, which contains about 2 million data points.
> > >
> > > We are open for further suggestions and discussions.
> > >
> > > - **_“the appendix suggests that the results in the paper also only consider the Jacobian of the last few layers of the network, rather than the whole DNN.”_**
> > >
> > > For deployments on a real robot, we find that using the Jacobian associated with the last few layers can be a practical alternative. To back up, for real robot experiments, we use the Jacobian of the last few layers while we still report the results where we use the Jacobian of the entire neural network in the appendix. In our formulation, this corresponds to pruning all the other layers except for the Jacobian of the last few layers. We find that by using the Jacobian of the last few layers, we can still perform novelty detection better than the widely used technique called MC-dropout, while being about 12 times faster on our hardware. The performance is better by considering the Jacobian of the entire neural network, but this results in slower inference time. Therefore, using the Jacobian of the last few layers can be a practical alternative in terms of run-time and performance trade-offs.
> > >
> > > In practice, other techniques such as MC-dropout may also use dropout associated with only few layers [2,3], while deep ensemble may keep only few ensemble members (as the reviewer points out) despite the potential deterioration of performance. Similarly, we argue that our approach can also use the Jacobian associated with only the last few layers, if it offers the practicality for the given use-case.
> > >
> > > We are open for further suggestions and discussions.

---

> > > > ### Author Response · Authors · 2021-08-26
> > > > **Response to Reviewer hvPu (4/4)**
> > > >
> > > >
> > > > **On thoroughness of the experiments:**
> > > >
> > > > - **_“Experiments comparing to Deep Ensembles are crucial, as they are a commonly used tool for DNN uncertainty, and can give good results even with a low number of ensemble members, allowing for efficient real-time application.”_**
> > > >
> > > > - **_AND “The main missing experiments are comparisons to Deep Ensembles.”_**
> > > >
> > > > We have included deep ensembles as an additional baselines in the revised manuscript. Please find the results in table 1. The implementation details are provided in the appendix. We thank the reviewer for the useful suggestion.
> > > >
> > > > On the efficiency of deep ensembles, we argue that the real-time capability of deep ensemble primarily depends on the application domains. For example, in robot perception, lowering the number of ensemble members may not allow for real-time applications due to the relatively larger size of the used neural networks. To expand our stance, lets consider an example of a robot perception task, where the robot is equipped with 8GB of memory (e.g. a Jetson TX2) and a single neural network is 5GB in GPU memory. In this case, lowering the number of ensembles cannot allow for real-time applications. Even in the case where a single neural network is 2GB and we can fit 4 ensemble members to a single GPU, paralleling the predictions on the real robot may not be trivially done – or we are not aware of any existing implementations or proof that this is possible for the real-robot object detection. The new research line on “sampling free” methods is attempting to address this limitation [1], and the recent survey paper on probabilistic object detection [2] also states this as an important direction of research.
> > > >
> > > > We would appreciate if the reviewer could enlighten us on this comment.
> > > >
> > > > [1] Sampling-free Epistemic Uncertainty Estimation Using Approximated Variance Propagation, Postels et al, ICCV 2019.
> > > >
> > > > [2] A Review and Comparative Study on Probabilistic Object Detection in Autonomous Driving, Feng et al, 2021.
> > > >
> > > > - **_“Furthermore, there are recent works considering different Laplace approximations that are closely related to a weight-space view of the DNN GP, that are worth discussing if not comparing against [1,2].”_**
> > > >
> > > > We have included the discussions of the mentioned papers in the revised manuscript (lines 290-300). Indeed, both the papers clearly showcase the relevance of the weight space view, and are closely linked to our work: our paper takes the direction of the function space view (e.g. GPs) while the mentioned papers focus on the weight space views (e.g. Bayesian linear regression).
> > > >
> > > > However, we note that Sharma et al [2] have been presented at UAI 2021, which was held about 1 month after the CoRL submission deadline. The weight space view of the NLMs (e.g. Madras et al [1]) is a well established topic (in agreements with reviewer Qn76), making it difficult to discuss or compare to all the prior works.
> > > >
> > > > We thank the reviewer for pointing out, which made our discussions on the prior works more complete.
> > > >
> > > > [1] Madras et al., “Detecting Extrapolation with Local Ensembles,” ICLR 2019
> > > >
> > > > [2] Sharma et al., “Sketching Curvature for Efficient Out-of-Distribution Detection for Deep Neural Networks,” UAI 2021

---

> > > > > ### Comment · Reviewer_hvPu · 2021-09-04
> > > > > **Thanks for your detailed comments**
> > > > >
> > > > > Thanks for your comments and highlighted revisions to the manuscript.
> > > > >
> > > > > With regard to the NLM / GP mismatch, the way the NLM was presented considered a an arbitrary loss function, for which the Hessian is not guaranteed to be input independent. Indeed, for commonly used loss functions, the hessian is input dependent, and I believe the updated manuscript makes this clearer. (though there appears to be a typo: "via a linearization around the DNNs’ last layer [24]. " -- the linearization is not limited to the last layer, but rather it is performed around the final trained weights)
> > > > >
> > > > > I still am a little confused by the notation $\sigma_{0,m}$ -- to my understanding, this should be equal to the variance of $\epsilon$, which according to the NLM correspondence, is equal to the constant value of $H_L^{-1}$.
> > > > > I can understand that in practice, it might be useful to allow this parameter to be learned from data, and learned independently for each mixture element, but to me this is a deviation from the NLM theory -- the connection between $\sigma_{0,m}$ and the loss hessian should be more clearly stated, and the choice to learn an independent value for the aleatoric noise per mixture element should be made clear.
> > > > >
> > > > > Thanks for taking the time to address my comments and incorporate my feedback -- I will increase my score.

---

### Official Review · Reviewer_mLeS · 2021-07-24

**Originality:** Very Good
**Technical Quality:** Very Good
**Clarity Of Presentation:** Good
**Impact:** 4

**Recommendation:**

Weak Accept: I recommend accepting the paper, but will not argue for my recommendation if the majority of other reviewers have a different opinion.

**Summary:**

This paper addresses the problem of introspection, an intrinsic understanding of how confident the model is in its own predictions. The paper provides a theoretical connection between DNNs and MoE-GPs and finally provides a practical learning application. Experiments are carried out on data sets derived from robotic sampling and inverse dynamics problems.


**Issues:**

Issues
- I would recommend improvements in the technical exposition of the material. For example, Section 3 can provide a roadmap for how the proof will develop.
- Does the current system has a way of forgetting points. In an online application, the incoming data may exceed available memory. Hence, is there a possibility for a principled approach for removing points as is done in Informative Vector Machines (IVMs)

**Reviewer Expertise:**

Good: General knowledge of the area

**Strengths And Weaknesses:**

Strengths
- The paper addresses an important problem of uncertainty estimation in prediction. The work provides a theoretical foundation from a Bayesian DNN and GPs viewpoint which culminates in a practical learning algorithm.
- The paper provides a scalable approach using a mixture of GPs approach and other techniques to reduce the online inference time making them suitable for robotics application.
Results on real robotics data sets.

Weaknesses
- The approach in the current form does not distinguish aleatoric and epistemic uncertainty in predictions and hence is likely to be less resilient to model uncertainty.


**Summary Of Recommendation:**

This paper contributes a theoretical framework for uncertainty estimation as well as a practical learning algorithm. The result is likely to have impact on real robotics applications.

---

> ### Author Response · Authors · 2021-08-26
> **Response to Reviewer mLeS**
>
>
> We thank the reviewer for the valuable time and feedback. Our line-by-line responses to the reviewer are provided below.
>
> - **_On “The approach in the current form does not distinguish aleatoric and epistemic uncertainty in the predictions and hence is likely to be less resilient to model uncertainty.”_**
>
> The predictive uncertainty of Gaussian Processes (GPs) refers to the combination of both aleatoric and epistemic uncertainty [1,2]. We have clarified this in the revised manuscript (lines 135-139 highlighted in blue).
>
> [1] Aleatoric and epistemic uncertainty in machine learning: An introduction to concepts and methods, Hüllermeier et al, Machine Learning 2021.
>
> [2] (Chapter 2) Gaussian Processes for Machine Learning, Rasmussen et al. MIT Press 2006.
>
> - **_On “I would recommend improvements in the technical exposition of the material. For example, Section 3 can provide a roadmap for how the proof will develop”_**
>
> In section 3, we provide a roadmap for how the proof will develop (figure 2 caption, highlighted in blue).
>
> We thank the reviewer for the suggestions.
>
> - **_On “Does the current system has a way of forgetting points. In an online application, the incoming data may exceed available memory. Hence, is there a possibility for a principled approach for removing points as is done in Informative Vector Machines (IVMs)”_**
>
> The current system has a way of selecting the most informative points as is done in Informative Vector Machines (equivalently removing the points) [1]. The description of the corresponding mechanism can be found in lines 203-206 (highlighted in blue). As pointed out, the available data may exceed available memory. Thus, our method relies on the ideas of informative vector machines to select fewer points, along with other mechanisms such as pruning and division of the data into smaller subsets.
>
> [1] Fast sparse Gaussian process methods: The informative vector machine, Lawrence et al, NeurIPs 2003.

---

### Author Response · Authors · 2021-08-26
**A Summary of the Revision**

We would like to thank the area chairs and the reviewers for their thoughtful and detailed comments, which improved our paper significantly.

Our work has been generally acknowledged as "important", "relevant" and "novel", and all the reviewers appreciated the results on a real robot experiment. Moreover, Reviewer mLeS and hvPU appreciated both the theoretic and the practical aspect of our work, and also our efforts in making the proposed approach suitable for many robotic applications. Reviewer hvPu further provided us a valuable opinion that the ideas have been communicated well, while Reviewer Q7n6 acknowledged the usefulness of our focus on "fast predictive models with useful uncertainty quantification" and indicated our contribution as trendy.

On the other hand, the reviewers generally raised legitimate concerns on the clarity of the presentation in section 3 (incl. design choices), and provided us their thoughtful suggestions for improvements. The reviewers also fairly suggested specific baselines for further comparisons and discussions, and urged for the clarifications regarding the scalability of the approach. More concerns have been also raised by the individual reviewers such as the need for distinction between model and data uncertainty, the need for the highlighted discussions of our limitations, and further clarifications in making the design choices more coherent.

In the light of the reviews, we have revised our manuscript and provided the requested experimental data. The list of the revisions are:

- **Reviewer mLeS:** the revision distinguishes between the model and the data uncertainty (lines 135-137).
- **Reviewer mLeS:** the revision provides a roadmap for how the proof will develop (caption of figure 2).
- **Reviewer mLeS:** the revision highlights the mechanisms of the IVMs (lines 203-204).
- **Reviewer hvPu:** the revision provides the intuition behind the modelling assumptions (in particular, the choice of covariance) (lines 95-109).
- **Reviewer hvPu:** the revision in lines 135-139 clarifies the modelling assumptions (in particular, the observation noise).
- **Reviewer hvPu:** the revision clarifies the connection between the regression uncertainty to a classification task (lines 146-151) .
- **Reviewer hvPu:** in the revision, figure 2 has been redesigned for clearer presentation of the theoretic results.
- **Reviewer hvPu:** the revision provides more details on the active learning and pruning strategies  (lines 641-50 of the appendix).
- **Reviewer hvPu:** the revision identifies the potential terms that could cause the confusion on scalability, and more precise statements have been provided.
- **Reviewer hvPu:** the revision includes Deep Ensembles as an additional baseline (table 1).
- **Reviewer hvPu:** the revision includes the suggested references on the weight-space view of the NLMs (lines 295-297).
- **Reviewer Q7n6:** the revision highlights and modifies the motivation and the validations of our design choices in green.
- **Reviewer Q7n6:** the revision contains the clarifications and the re-writings on the NLMs and the used GPs. The mentioned references have been added.
- **Reviewer Q7n6:** the revision clarifies the memory requirement of a Jetson TX2 (caption of figure 6).
- **Reviewer Q7n6:** the revision contains a comparison to the standard GPs with the NTK and the RBF as the kernel functions (figure 2-3 of the appendix).
- **Reviewer Q7n6:** in the revision, the summary of the contributions have been revised to only include the major points (lines 43-46).
- **Reviewer Q7n6:** in the revision, the pointed typos have been corrected.

More detailed clarifications have been provided to the individual reviewers below. We remain open to further discussions.

---

### Author Response · Authors · 2021-09-27
**More links about this project.**

Followings are the relevant links to our paper (work-in-progress currently).

(Code) <https://github.com/DLR-RM>

(Video) <https://www.youtube.com/user/DLRRMC>

---

### Meta-Review · Area_Chair_Bsw7 · 2021-08-02

**Recommendation:** Accept (Poster)
**Confidence:** 4

**Metareview:**

This paper proposes an approach to novelty detection for deep neural networks using Gaussian Processes (GPs). More specifically, the paper uses mixture of experts (MoE)-GPs with a neural network-based kernel to estimate predictive uncertainty associated with a neural network’s output. The resulting approach is demonstrated on two robotic tasks corresponding to learning the inverse dynamics of a manipulator and object detection on a micro-aerial vehicle.

Strengths:
+ The reviewers are generally in agreement that the paper addresses an important problem (uncertainty estimation for DNNs) that is relevant to robotics.
+ The reviewers generally agree that there is novelty in the technical approach (using MoE-GPs to perform uncertainty estimation).
+ The reviewers appreciated the experimental results on real-world robotics datasets.

Weaknesses:
- The primary concern raised by the reviewers related to the technical presentation of the approach. In particular, the theoretical derivations in Section 3 are not clearly described; certain technical choices are not clearly motivated or explained (see the comments from Reviewers mLeS and hvPu for detailed comments and suggestions).
- There are also concerns regarding the scalability of the approach. The onboard robotic experiments use relatively small datasets and the Jacobian used for uncertainty estimation is only associated with the last few layers of the DNN. Reviewers also raised concerns regarding the memory footprint of the approach.
- Reviewers suggested additional comparisons with baselines, including Deep Ensembles, and GPs.

Suggestions:
I urge the authors to consider the reviewers’ detailed feedback in order to improve the paper. The primary suggestions for improvement include:
- Improving the technical exposition and justifying theoretical derivations and choices in Section 3.
- Addressing the concerns regarding scalability of the approach.
- Providing additional comparisons with baselines (and including discussions of some of the papers that the reviewers mentioned; particularly the ones mentioned by Reviewer Q7n6).

----- Post rebuttal -----

The reviewers have made significant improvements to the paper in the rebuttal and revision period. In particular, they have addressed the primary concerns raised by reviewers by (i) adding an additional strong baseline (Deep Ensembles), (ii) addressing questions regarding the memory footprint, and (iii) improving the presentation of the work. Given the strengths of the work in terms of importance of the problem, novelty, and real-world robotics evaluations (and also taking into account the improvements made to the paper), I believe this work represents a strong contribution.

---

> ### Author Response · Authors · 2021-08-26
> **Response to the Area Chairs (1/2)**
>
>
> We thank the area chairs for their valuable time and feedback. We address the raised concerns below and point out the improvements of our manuscript in the light of the reviews.
>
> **On the technical exposition:**
>
> We have incorporated all the suggestions made by the reviewers (highlighted in blue within the revised manuscript). Moreover, we have revised the justification of theoretical derivations and their choices (highlighted in green within the revised manuscript).
>
> **On additional comparisons with baselines:**
>
> - **_“Reviewers suggested additional comparisons with baselines including Deep Ensemble, and GPs”_**
>
> We provide the suggested baselines including deep ensembles (table 1 in the main manuscript) and 2 full GPs with the RBF kernel and the neural network-based kernel (figure 2 and 3 in the appendix).
>
> - **_“including discussions of some of the papers that the reviewers mentioned; particularly the ones mentioned by Reviewer Q7n6”_**
>
> We provide the discussions of the papers mentioned by the reviewers. In particular to reviewer Q7n6:
>
> First, the reviewer states that “the neural linear model is actually an established class of Bayesian Neural Networks [1,2], so you need a different term”. However, the neural linear model that the reviewer is referring to, is the same class of model that we define in lines 95-109. We address these points to reviewer Q7n6 in more detail and revised our manuscript accordingly.
>
> Second, the reviewer states that “The paper does not cite [3], which is a GP version of the idea presented here”. However, the referred paper [3] is not the GP version of the idea presented in our approach. We address these points to reviewer Q7n6 by discussing the differences and the similarities, and revised our manuscript accordingly.
>
> [1] Benchmarking the Neural Linear Model, Ober et al, AABI 2020.
>
> [2] Neural Linear Models with Gaussian Process Priors, Watson et al, AABI 2020.
>
> [3] Incremental Local Gaussian Regression, Meier et al, NeurIPs 2014.

---

> > ### Author Response · Authors · 2021-08-26
> > **Response to the Area Chairs (2/2)**
> >
> >
> > **On the concerns regarding the scalability:**
> >
> > - **_“The onboard robotic experiments use relatively small datasets”_**
> >
> > The purpose of real robot experiments is to validate if our approach can work on a physical robot, not to test the scalability. To explain, we had to perform the flight experiments to collect the data, and manually annotate the images for transfer learning. We stopped creating more labeled data once we could realize the demonstration (seen in the attached video). As creating relatively larger labeled dataset is both not practical and required in our scenario, we decided to test the scalability of our approach in other experiments, in particular, by using the KUKA simulation dataset (about 2 million data points as evidenced in figure 4).
> >
> > We have provided the corresponding discussions to the reviewers and revised our manuscript accordingly.
> >
> > - **_“The Jacobian used for uncertainty estimation is only associated with the last few layers.”_**
> >
> > For deployments on a real robot, the Jacobian associated with the last few layers can be a practical alternative. To back up, we compare the use of the Jacobian associated with the last few layers by reporting the results where we use the Jacobian of the entire neural network (figure 5 in the appendix). In our formulation, this corresponds to pruning all the other layers except for the Jacobian of the last few layers. We find that by using the Jacobian of the last few layers, we can still perform novelty detection better than the widely used technique called MC-dropout [1], while being about 12 times faster on our hardware (figure 6). The performance improves by considering the Jacobian of the entire neural network along with pruning, but this resulted in slower inference time (still faster than our baseline: MC-dropout). Therefore, we find that using the Jacobian of the last few layers can be a practical alternative in terms of run-time and performance trade-offs.
> >
> > In practice, other techniques such as MC-dropout may use dropout associated with only few layers (e.g [2]), while deep ensemble may keep only few ensemble members (as reviewer hvPu argues), despite the potential deterioration of performance. Similarly, we argue that our approach can also use the Jacobian associated with only the last few layers, if it offers the demonstrated practicality.
> >
> > We have provided the corresponding discussions to the reviewers.
> >
> > [1] Dropout as a Bayesian approximation: Representing model uncertainty in deep learning, Gal et al, ICML 2016.
> >
> > [2] Introspective robot perception using smoothed predictions from Bayesian neural networks, Feng et al, ISRR 2019.
> >
> >
> > - **_“Reviewers also raised concerns regarding the memory footprint of the approach.”_**
> >
> > We acknowledge that as we build on Bayesian non-parametric approaches such as GPs, our method can require relatively large amounts of memory that scales with the number of data points. We have addresses this point in the revised manuscript by taking the suggestions of the reviewers, and moving the previous discussions on this point from the appendix to the main text.
> >
> > The requested data on the memory footprint of our method and other approaches are provided to reviewer Q7n6. In our real-world scenario specifically, we provide the data to show that the mentioned alternatives such as deep ensembles require more memory than our approach, while we acknowledge that in the other use-cases beyond our consideration, deep ensemble could require less memory.
> >
> > - **“_Addressing the concerns regarding the scalability of the approach”._**
> >
> > To further address the comments on scalability, we argue that our specific claims on scalability have been substantiated. To explain, the paper claimed that our approach “offers an improved scalability when compared to a full Gaussian Process (GP) with a neural network-based kernel” (line 36-37), and “show that our method scales to approximately 2 million data points for the task of learning inverse dynamics (line 40-41)”. Our experiments validate these claims. For the former, our benchmark scenario (table 1) involves the regime of data where a standard GPs do not scale (literature refers to big dataset if more than 10000 data points because a full GP does not scale [1, 2]). Figure 4 supports the later claim by measuring the time to train on the 2 million data points. Thus, our experiments back up our specific claims on scalability, which are to show the advancements in the scalability of GPs with a neural network-based kernel.
> >
> > As we acknowledge the potential confusion of the readers, we have revised our manuscript accordingly.
> >
> > [1] When Gaussian Process Meet Big Data: A Review of Scalable GPs, Liu et al, IEEE transactions on neural networks and learning systems, 2020.
> >
> > [2] (Chapter 8) Gaussian Processes for Machine Learning, Rasmussen et al. MIT Press 2006.

---

### Decision · Program_Chairs · 2021-09-13

**Decision:**

Accept (Poster)

**Comment:**

This paper proposes an approach to novelty detection for deep neural networks using Gaussian Processes (GPs). More specifically, the paper uses mixture of experts (MoE)-GPs with a neural network-based kernel to estimate predictive uncertainty associated with a neural network’s output. The resulting approach is demonstrated on two robotic tasks corresponding to learning the inverse dynamics of a manipulator and object detection on a micro-aerial vehicle.

Strengths:
+ The reviewers are generally in agreement that the paper addresses an important problem (uncertainty estimation for DNNs) that is relevant to robotics.
+ The reviewers generally agree that there is novelty in the technical approach (using MoE-GPs to perform uncertainty estimation).
+ The reviewers appreciated the experimental results on real-world robotics datasets.

Weaknesses:
- The primary concern raised by the reviewers related to the technical presentation of the approach. In particular, the theoretical derivations in Section 3 are not clearly described; certain technical choices are not clearly motivated or explained (see the comments from Reviewers mLeS and hvPu for detailed comments and suggestions).
- There are also concerns regarding the scalability of the approach. The onboard robotic experiments use relatively small datasets and the Jacobian used for uncertainty estimation is only associated with the last few layers of the DNN. Reviewers also raised concerns regarding the memory footprint of the approach.
- Reviewers suggested additional comparisons with baselines, including Deep Ensembles, and GPs.

Suggestions:
I urge the authors to consider the reviewers’ detailed feedback in order to improve the paper. The primary suggestions for improvement include:
- Improving the technical exposition and justifying theoretical derivations and choices in Section 3.
- Addressing the concerns regarding scalability of the approach.
- Providing additional comparisons with baselines (and including discussions of some of the papers that the reviewers mentioned; particularly the ones mentioned by Reviewer Q7n6).

----- Post rebuttal -----

The reviewers have made significant improvements to the paper in the rebuttal and revision period. In particular, they have addressed the primary concerns raised by reviewers by (i) adding an additional strong baseline (Deep Ensembles), (ii) addressing questions regarding the memory footprint, and (iii) improving the presentation of the work. Given the strengths of the work in terms of importance of the problem, novelty, and real-world robotics evaluations (and also taking into account the improvements made to the paper), I believe this work represents a strong contribution.